# Hsp90 provides a platform for kinase dephosphorylation by PP5

Maru Jaime-Garza[1], Carlos A. Nowotny [1], Daniel Coutandin[2], Feng Wang[1], Mariano Tabios[1] & David A. Agard [1] ✉

The Hsp90 molecular chaperone collaborates with the phosphorylated Cdc37 cochaperone for the folding and activation of its many client kinases. As with many kinases, the Hsp90 client kinase CRaf is activated by phosphorylation at specific regulatory sites. The cochaperone phosphatase PP5 dephosphorylates CRaf and Cdc37 in an Hsp90-dependent manner. Although dephosphorylating Cdc37 has been proposed as a mechanism for releasing Hsp90-bound kinases, here we show that Hsp90 bound kinases sterically inhibit Cdc37 dephosphorylation indicating kinase release must occur before Cdc37 dephosphorylation. Our cryo-EM structure of PP5 in complex with Hsp90:Cdc37:CRaf reveals how Hsp90 both activates PP5 and scaffolds its association with the bound CRaf to dephosphorylate phosphorylation sites neighboring the kinase domain. Thus, we directly show how Hsp90's role in maintaining protein homeostasis goes beyond folding and activation to include post translationally modifying its client kinases.

Maintaining protein homeostasis is a critical function for all organisms and relies on a broad array of proteins including molecular chaperones[1]. The Heat shock protein Hsp90 is a molecular chaperone required for the folding and activation of over 10% of the human proteome[2,3]. Hsp90's "client" proteins are enriched in signaling proteins such as protein kinases, transcription factors, and steroid hormone receptors. This leads Hsp90 to play an important role in organismal health and disease. Importantly, more than half of all human kinases depend on Hsp90 and the Hsp90 cochaperone Cdc37 for their folding and activation[4]. The role of Hsp90 in kinase activation goes beyond folding and includes facilitating alterations in post-translational modifications. Through its regulation of both kinase folding and kinase dephosphorylation, Hsp90 can modulate essential signaling pathways.

One such critical Hsp90-dependent pathway is the Ras-MAPK pathway involved in regulating cellular proliferation[5,6]. When dysregulated, this pathway is often implicated in cellular malignancy[7]. Raf kinases are part of this pathway and act to propagate growth hormone signals from the membrane-bound Ras GTPase to MEK and ERK kinases which can lead to signal amplification[8]. Pathway activation requires Raf kinase dimerization which is mediated by phosphorylation of the Raf kinase acidic N-terminus[9–11]. Thus, dephosphorylation of this acidic N-terminus may then lead to pathway inactivation[12].

While Raf kinase activation has been extensively studied, Raf inactivation is less well understood. Importantly, for RAF proto-oncogene serine/threonine-protein kinase (CRaf or RAF-1), a member of the Raf family, the Hsp90 cochaperone, Serine/threonine-protein phosphatase 5 (PP5) has been directly implicated in its dephosphorylation and inactivation[13]. More specifically, PP5 was shown to pulldown with CRaf and specifically dephosphorylate phosphoserine 338 (CRaf$^{pS338}$) during Ras-MAPK pathway activation. Similarly, siRNA PP5 knockdown led to a specific increase in S338 phosphorylation. Based on these results, PP5 was hypothesized to play a key role in CRaf inactivation.

PP5 is a serine-threonine phosphatase from the PPP family, which consists of a Tetratricopeptide (TPR) domain N-terminal to the catalytic phosphatase domain[14,15]. The TPR domain sits directly atop the catalytic domain, sterically blocking substrate binding and access to the active site[16,17]. The inhibitory αJ helix on the catalytic domain stabilizes the autoinhibited PP5 state through hydrophobic interactions with the TPR domain[18]. Like other TPR-cochaperones, the PP5 TPR domain binds the Hsp90 C-terminal MEEVD tail, in this case leading to

[1]Department of Biochemistry and Biophysics, University of California, San Francisco, San Francisco, CA 94143, USA. [2]Novartis Institutes for BioMedical Research, San Diego, CA 92121, USA. ✉e-mail: agard@msg.ucsf.edu

PP5 activation[16,19]. A TPR domain mutation that blocks MEEVD binding abrogates the dephosphorylation of CRaf[pS338] and inhibits PP5 coelution with CRaf kinase[13]. These results strongly suggest that Hsp90 plays a key role in CRaf dephosphorylation by controlling when and where PP5 becomes activated. In addition to CRaf, PP5 dephosphorylates numerous other Hsp90 clients, presumably while they are bound to Hsp90[20–25].

Ppt1, the yeast homolog of PP5 has been shown to dephosphorylate yeast Hsp90 itself[26]. This led Vaughan et al. to hypothesize that PP5 might dephosphorylate the Hsp90 cochaperone Cdc37, which must be phosphorylated on S13 (Cdc37[pS13]) to function in kinase activation[27–30]. Cdc37 is a kinase-specific Hsp90 cochaperone that binds and destabilizes kinase domains, enabling their recruitment into Hsp90:Cdc37:kinase complexes[31]. Hsp90:Cdc37:kinase complexes purified from yeast, baculovirus, or mammalian cells are invariably phosphorylated on Cdc37[S13] [29,32]. It has also been shown that the mutation of Cdc37[S13] leads to a decrease in Hsp90 and kinase pulldowns[28,29]. Finally, structural analysis of Hsp90:Cdc37:kinase complexes reveals that Cdc37[pS13] is required to stabilize the Cdc37 N-terminal domain and to facilitate interactions with Hsp90[33].

Without Hsp90 activation, PP5 by itself cannot dephosphorylate isolated Cdc37 or Cdc37 within a Cdc37:Cdk4 complex[32]. PP5 can, however, dephosphorylate Cdc37 in the context of a purified Hsp90:Cdc37:Cdk4 complex, whereas non-specific phosphatases cannot[32]. This led to the hypothesis that PP5 must dephosphorylate Cdc37 while it is bound to an Hsp90:Cdc37:kinase complex and that it can thus facilitate kinase modification or release from the Hsp90 complex.

Here, to further understand the molecular mechanisms by which PP5 is activated and selects its target substrates, we determine the atomic resolution cryo-EM structure of a human Hsp90:Cdc37:CRaf:PP5 complex and biochemically explore PP5-dependent dephosphorylation of both CRaf and Cdc37. Surprisingly, this reveals that while CRaf can be readily dephosphorylated by Hsp90-activated PP5, Cdc37 can only be dephosphorylated by Hsp90-activated PP5 in the absence of bound kinase. Through this work, we propose a mechanism for PP5 activation; we suggest that PP5 does not serve as a kinase release factor but instead blocks kinase rebinding to previously accessed Hsp90:Cdc37 complexes.

## Results

### Only kinase-free Hsp90-Cdc37 complex can be dephosphorylated by PP5

PP5 is reported to specifically dephosphorylate CRaf[pS338] in vivo while leaving other essential CRaf phosphosites unaltered. Phosphorylated CRaf can be purified in Hsp90:Cdc37:CRaf complexes, and CRaf[pS338] can subsequently be dephosphorylated in vitro by PP5[34]. Unless otherwise noted, truncated CRaf[304–648] complexes (Hsp90:Cdc37:CRaf[304–648]) were used to measure PP5-driven dephosphorylation. The phosphorylation state of CRaf kinase was quantitatively assessed using specific phosphosite antibodies directed against CRaf[pS338](N-terminal to the kinase domain), and the control phosphosite CRaf[pS621] (C-terminal to the kinase domain) (Supplementary Fig. 1). In addition, Cdc37 dephosphorylation was probed using Cdc37[pS13] specific antibodies[13].

PP5 activity was followed by incubating *Escherichia coli* purified PP5 with mammalian purified Hsp90:Cdc37:CRaf complexes (Fig. 1a and Supplementary Fig. 2). CRaf[pS338] was promptly dephosphorylated. Surprisingly, the CRaf[pS621] control site was also rapidly dephosphorylated, although at about 40% the rate of CRaf[pS338]. Contrary to our expectations, Cdc37[pS13] was not measurably dephosphorylated upon PP5 addition. These results corroborate structural data showing that Cdc37[pS13] is inaccessible within the complex, yet contradict previous in vitro biochemical experiments[33]. With substantially longer incubations at 37 °C (vs RT) we can observe Cdc37[pS13]

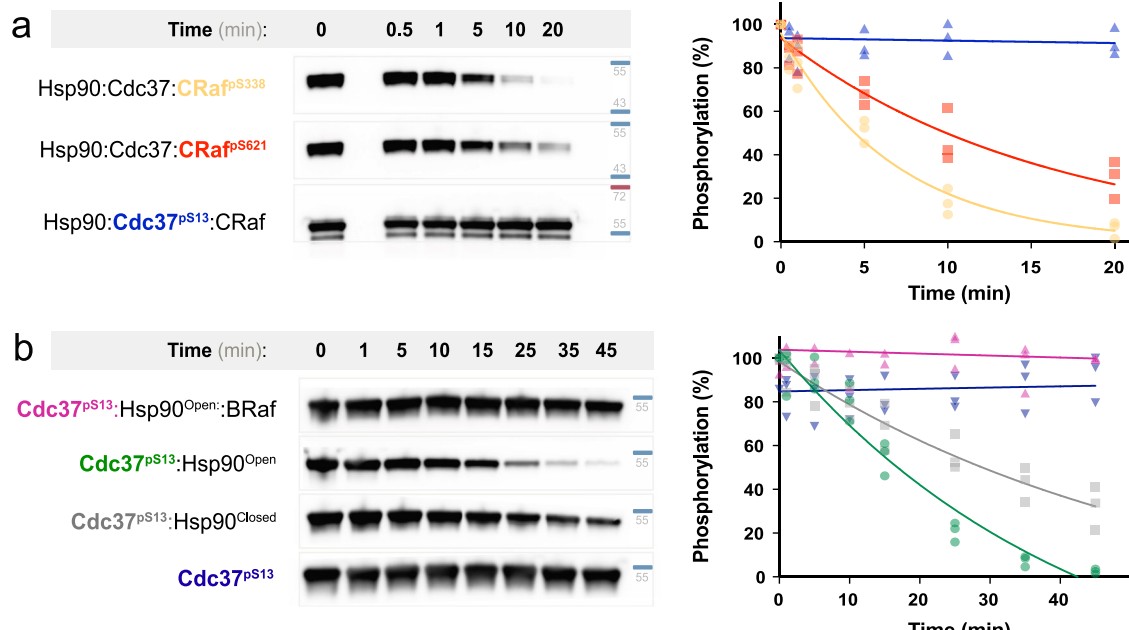

**Fig. 1 | Kinase sterically blocks Cdc37 dephosphorylation. a** Mammalian purified Hsp90:Cdc37:CRaf complex (1.5 mM) was incubated with PP5 (75 nM) at 25 °C. The dephosphorylation of CRaf[pS338], CRaf[pS621] and Cdc37[pS13] was assayed by phospho-specific blotting (*n* = 3, rate ± SEM). CRaf[pS338] was preferentially dephosphorylated (yellow, 0.147 ± 0.012 min⁻¹), CRaf[pS621] was more slowly dephosphorylated (red, 0.063 ± 0.006 min⁻¹), while Cdc37[pS13] dephosphorylation was not apparent (blue, 0.001 ± 0.002 min⁻¹). Both native and tagged Cdc37 are visible in the α-Cdc37[pS13] blot. **b** Cdc37[pS13] (3 mM) was incubated with equimolar complex components (Hsp90[open dimer], Hsp90[closed dimer], BRaf) and PP5 (750 nM) at 25 °C (*n* = 3, rate ± SEM). Cdc37[pS13] dephosphorylation was assayed by α-Cdc37[pS13] blotting. There is no dephosphorylation when BRaf kinase is added to the Hsp90[open]:Cdc37[pS13] complex (magenta, 0 min⁻¹). PP5 dephosphorylates Cdc37[pS13] bound to Hsp90[open] (green, 0.024 ± 0.009 min⁻¹) faster than when bound to Hsp90[closed] (gray, 0.018 ± 0.01 min⁻¹). PP5 does not dephosphorylate Cdc37[pS13] when Hsp90 is absent (blue, 0.004 ± 0.37 min⁻¹). All western blot data is available in Supplementary Information, and quantified data available in the provided Source Data file.

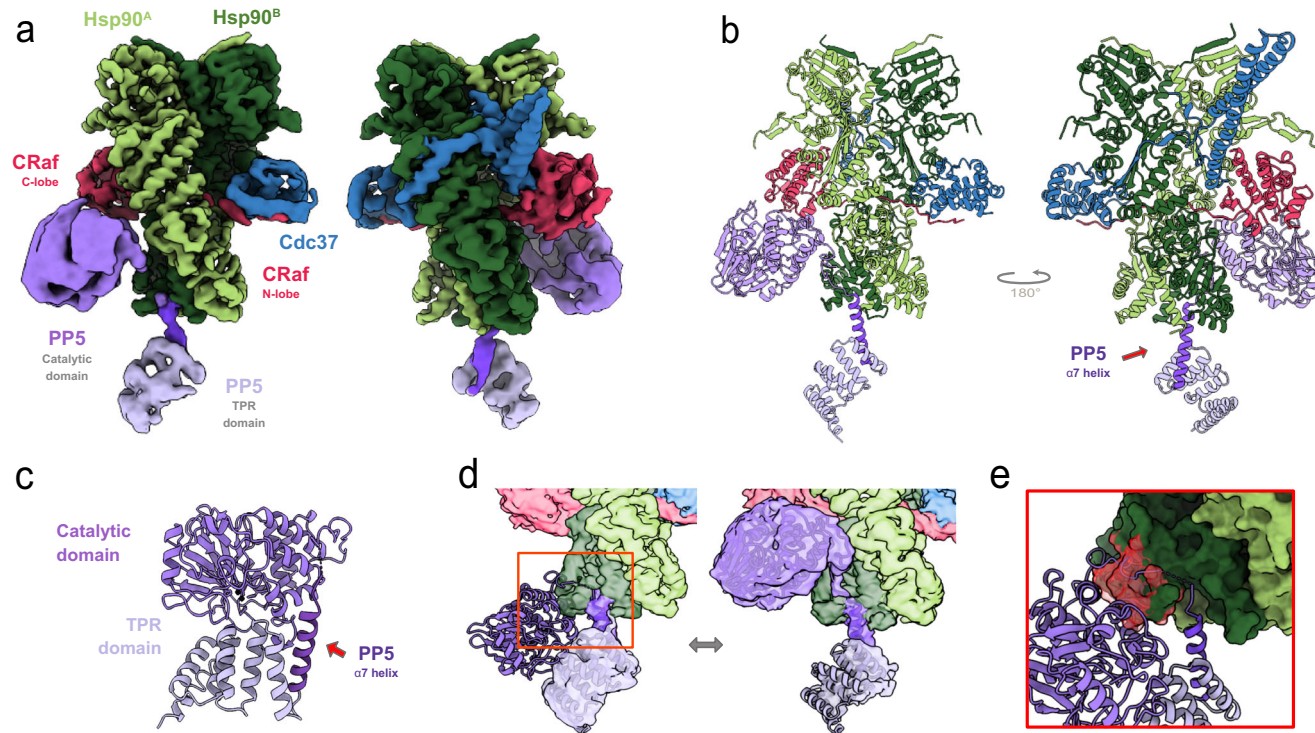

**Fig. 2 | Hsp90 activates PP5, acting as a phosphatase scaffold. a** Cryo-EM composite map and Hsp90:Cdc37:CRaf:PP5 complex model (**b**) show unfolded CRaf kinase (magenta) threaded through the closed Hsp90 dimer lumen (green), and embraced by Cdc37 (blue). PP5 is in an active conformation as its TPR domain (lavender) interacts with the Hsp90 CTDs, while the PP5 catalytic domain (purple) is near the Hsp90[MD] and the CRaf C-lobe. **c** In the PP5 crystal structure (PDB: [1WAO]), the PP5 active site is occluded when not bound to Hsp90. **d** Overlaying the auto-inhibited PP5 structure with the activated structure reveals PP5-Hsp90 steric clashes (**e**, in red) that would require rearrangement of the PP5 domain interface, possibly driving PP5 activation.

dephosphorylation. However, size exclusion analysis suggests that this is due to partial complex dissociation at 37 °C (Supplementary Fig 3).

To probe the factors contributing to the inaccessibility of Cdc37[pS13], we systematically explored the impact of the Hsp90 nucleotide state and the role that the kinase plays in Cdc37[pS13] dephosphorylation. As shown by cryo-EM, Hsp90 appears closed in the natively isolated Hsp90:Cdc37:kinase complexes, leaving Cdc37[pS13] inaccessible[33]. In principle, by leaving out the nucleotide in an in vitro reconstitution (Hsp90[open]), it should be possible to form Hsp90 open-state complexes. Unfortunately, it has not yet been possible to reconstitute CRaf assembly into an Hsp90 complex in vitro. However, it is possible to assemble Hsp90:kinase domain complexes using the heavily mutated and solubilized BRaf kinase domain[35,36]. Hsp90, Cdc37, and BRaf were expressed and purified from *E. coli*, and Cdc37 was phosphorylated by CK2 before a final purification. We reconstituted an Hsp90:Cdc37:BRaf complex by mixing and incubating for 30 min at 4 °C[35]. Dephosphorylation of Cdc37[pS13] upon addition of PP5 was again quantitated (Fig. 1b). To our surprise, PP5 was unable to dephosphorylate Cdc37[pS13] reconstituted into Hsp90[open] Hsp90:Cdc37:BRaf complexes.

To test if BRaf might sterically occlude PP5 from dephosphorylating Cdc37, we repeated the experiment without the kinase present. Notably, Hsp90-bound Cdc37[pS13] was rapidly dephosphorylated by PP5 without the presence of any nucleotide. We reasoned that if kinase steric hindrance were the main factor limiting Cdc37[pS13] dephosphorylation, then closing Hsp90 without BRaf present might allow PP5 access to Cdc37[pS13]. Hsp90 was incubated with AMPPNP and allowed to shift to a stabilized closed conformation as has been shown previously[37]. Once Hsp90 shifted to a closed conformation, Cdc37[pS13] was added and dephosphorylation was measured. Interestingly, Hsp90[closed]:Cdc37[pS13] was dephosphorylated at about half the rate as Hsp90[open]:Cdc37[pS13], suggesting that while the Hsp90

conformational state contributes to the occlusion of Cdc37[pS13], kinase presence or absence dominates PP5 activity on Cdc37[pS13].

## PP5 becomes activated and uses Hsp90 as a dephosphorylation scaffold

Motivated by the high levels of PP5 activity on CRaf, we set out to determine the atomic structure of PP5 dephosphorylation in action. Various steps were taken to reduce sample heterogeneity, and protect the complex from dissociation at the air-water interface: (1) following from previous cryo-EM studies, natively isolated Hsp90[closed]:Cdc37:-CRaf complexes containing just the CRaf kinase domain (CRaf[KD], residues 336–618) were used, (2) the catalytically inactive PP5 mutant PP5[H304A] was used to stabilize substrate-bound PP5 without dephosphorylating CRaf[13,17], and (3) the complex was chemically crosslinked and frozen on grids covered with a monolayer of graphene oxide derivatized with amino-PEG groups (Supplementary Fig. 4)[38,39]. A large single particle dataset was collected and processed using cryoSPARC and RELION[40,41].

Despite having a biochemically homogeneous crosslinked sample, data processing indicated a conformationally heterogeneous complex requiring significant 3D classification and refinement yielding a 3.8 Å map (Fig. 2a and Table 1). In this map, Hsp90 si-dechains could be readily interpreted and more detail for Cdc37 and the kinase was observed than in our previous, lower-resolution Hsp90:Cdc37:Cdk4 kinase complex structure[33]. As expected from the Cdk4 kinase complex, and further seen in the recent CRaf kinase complex[42], the CRaf[KD] kinase domain in our structure was split into its distinct lobes, threaded through the Hsp90 lumen and stabilized by Cdc37. This new map shows a closed Hsp90, with the CRaf[KD] C-lobe extending from the Hsp90 lumen on one side while the prominent Cdc37 N-terminal coiled-coil projects away from the opposite complex surface. In the refined CRaf map, we observed the β4-strand

**Table 1 | Cryo-EM data collection, refinement and validation statistics**

| | #1 Hsp90:Cdc37:CRaf:PP5 Conformation I (EMDB-29984) (PDB 8GFT) | #2 Hsp90:Cdc37:CRaf:PP5 Conformation II (EMDB-29895) (PDB 8GAE) |
|---|---|---|
| *Data collection and processing* | | |
| Magnification | 105,000 | 105,000 |
| Voltage (kV) | 300 | 300 |
| Electron exposure (e⁻/Å²) | 69.0 | 69.0 |
| Defocus range (µm) | −0.8 to –1.8 | −0.8 to –1.8 |
| Pixel size (Å) | 0.835 | 0.835 |
| Symmetry imposed | No | No |
| Initial particle images (*n*) | 3,730,385 | 3,730,385 |
| Final particle images (*n*) | 545,237 | 522,000 |
| Map resolution (Å) | 3.8 | 3.3 |
| FSC threshold | 0.143 | 0.143 |
| Map resolution range (Å) | 2.9 to 7.5 | 2.9 to 7.5 |
| *Refinement* | | |
| Initial model used (PDB code) | 5fwk | 5fwk |
| Model resolution (Å) | 3.02 | 2.72 |
| FSC threshold | 0.143 | 0.143 |
| Model resolution range (Å) | | |
| Map sharpening *B* factor (Å²) | | |
| Model composition | | |
| Non-hydrogen atoms | 17,801 | 17,748 |
| Protein residues | 2181 | 2175 |
| Ligands | 8 | 8 |
| *B* factors (Å²) | | |
| Protein | 202.95 | 217.56 |
| Ligand | 126.34 | 124.30 |
| R.m.s. deviations | | |
| Bond lengths (Å) | 0.016 | 0.017 |
| Bond angles (°) | 2.112 | 1.956 |
| Validation | | |
| MolProbity score | 0.90 | 0.62 |
| Clashscore | 1.55 | 0.34 |
| Poor rotamers (%) | 0.05 | 0.1 |
| Ramachandran plot (%) | | |
| Favored | 98.01 | 98.47 |
| Allowed | 1.95 | 1.53 |
| Disallowed | 0.05 | 0.00 |

from the kinase N-lobe bound to a groove in the Cdc37$^{MD}$, as shown in recent work[42].

Beyond these features within the kinase complex, two new densities were visible. The first sits near the Hsp90 C-terminal domains (Hsp90$^{CTD}$) and the second reaches toward the CRaf C-lobe and the Hsp90 middle domain (Hsp90$^{MD}$) of protomer A. Focused local 3D classification around the newfound densities and subsequent refinement provided higher resolution views resulting in the composite map shown in Fig. 2a (Supplementary Fig. 5).

A complete atomic model (Fig. 2b) constructed using the consensus and composite maps clearly revealed a PP5 conformation quite distinct from that found in the autoinhibited crystal structure (Fig. 2c)[16,43]. Instead of sitting atop the PP5 catalytic domain active site,

the PP5 TPR domain was disassociated from its catalytic domain and interacted with both Hsp90$^{CTD}$s (Supplementary Movie 1). A low-resolution linker could be seen joining the two domains (Fig. 2d).

The PP5 TPR domain α7 helix made an end-on interaction with Hsp90, nestled into a groove between the two Hsp90$^{CTD}$s (Fig. 2d). This interaction leads to a modest widening of the Hsp90$^{CTD}$ groove as compared to the Hsp90:Cdc37:CRaf structure[42] (Supplementary Fig. 6). Fitting the TPR domain from the autoinhibited PP5 crystal structure into our Hsp90-bound structure revealed slight TPR domain rearrangements, but a substantial steric clash between the PP5 catalytic domain and the kinase if the catalytic domain were to remain in this inhibited position (Fig. 2e). Therefore, Hsp90 binding to the PP5 TPR domain likely prompts TPR domain-catalytic domain reorganization and potentially drives domain separation leading to the active PP5 conformation seen here.

Surprisingly, the PP5 catalytic domain was not located near the readily dephosphorylated CRaf$^{pS338}$ but was instead located on the opposite side of the Hsp90 dimer, with the PP5 active site facing the CRaf C-lobe. Although CRaf$^{617}$ and CRaf$^{618}$ are disordered in our focused maps, a CRaf low-resolution mainchain density can be traced near the PP5 active site in some classes (see below). This would place CRaf$^{pS621}$ near the PP5 active site, making pS621 readily accessible for PP5 dephosphorylation. Combined with our dephosphorylation data (Fig. 1a), this indicates that PP5 would be able to dephosphorylate CRaf$^{pS621}$ while bound to Hsp90.

**PP5's TPR domain interacts extensively with Hsp90's C-terminal domains via an extended helix**

In its activated conformation, the PP5 α7 helix bound in the amphipathic grove formed by Hsp90$^{CTD}$ helices from both protomers (Fig. 3a). Binding at this interface stabilized and elongated the C-terminal end of the α7 helix by a turn (Supplementary Fig. 7b). Interactions with Hsp90 are electrostatic superficially, and hydrophobic deep within the CTD groove (Fig. 3b, c).

The PP5 linker connecting the catalytic domain to the α7 helix has conserved acidic residues (PP5$^{D155}$ and PP5$^{E156}$) that lie in close proximity to Hsp90$^{R679}$ and Hsp90$^{R682}$ (Fig. 3c and Supplementary Fig. 7d), stabilizing the α7 helix within the Hsp90 amphipathic groove. Here, the PP5$^{F148}$ and PP5$^{I152}$ residues which were solvent exposed and disordered in the PP5 autoinhibited state (Supplementary Fig. 7a), were stabilized by Hsp90 hydrophobic residues (Hsp90$^{A}$: L638, L654, L657 and Hsp90$^{B}$: I680, I684, M683, L686) (Fig. 3d).

Density corresponding to the Hsp90$^{A}$ tail can be seen extending beyond the last Hsp90$^{CTD}$ α helix toward the PP5 TPR domain (Fig. 3d). While poorly resolved, the charge complementarity between the Hsp90$^{A}$ tail (Hsp90$^{D691-E694}$) and basic residues on the PP5 TPR domain (PP5$^{R150}$, PP5$^{R113}$, PP5$^{R117}$) suggest that Hsp90 and PP5 interact beyond the α7 helix (Fig. 3e). This electrostatic interaction may guide the Hsp90 tail toward the PP5 MEEVD binding site, for which heterogeneous density is visible in our map (Supplementary Fig. 7e).

The "entrance" for the TPR α7 helix on the Hsp90 CTD pyramidal groove measures approximately 13 Å wide when Hsp90 is in a closed position, but contracts by almost 3 Å when Hsp90 is in the Hsp90 partially open conformation found in the Hsp90:Hsp70:Hop:GR client loading state (Supplementary Fig. 6)[44]. This suggests that PP5 binding might be substantially weaker in the fully open apo state or the partially open client loading state.

To determine the functional significance of the PP5-Hsp90$^{CTD}$ interactions observed here, we mutated three key residues (PP5$^{R113D}$, PP5$^{F148K}$, PP5$^{R150D}$) and assessed their impact on both CRaf$^{pS338}$ and Hsp90$^{open}$:Cdc37$^{pS13}$ dephosphorylation (Fig. 3e and Supplementary Figs. 7a and 8). All three individual mutations led to ~3-fold reduction in dephosphorylation rate for CRaf$^{pS338}$ and a greater impairment for Hsp90$^{open}$:Cdc37$^{pS13}$. We next tested the binding of each PP5 mutant to increasing concentrations of closed Hsp90 dimers through

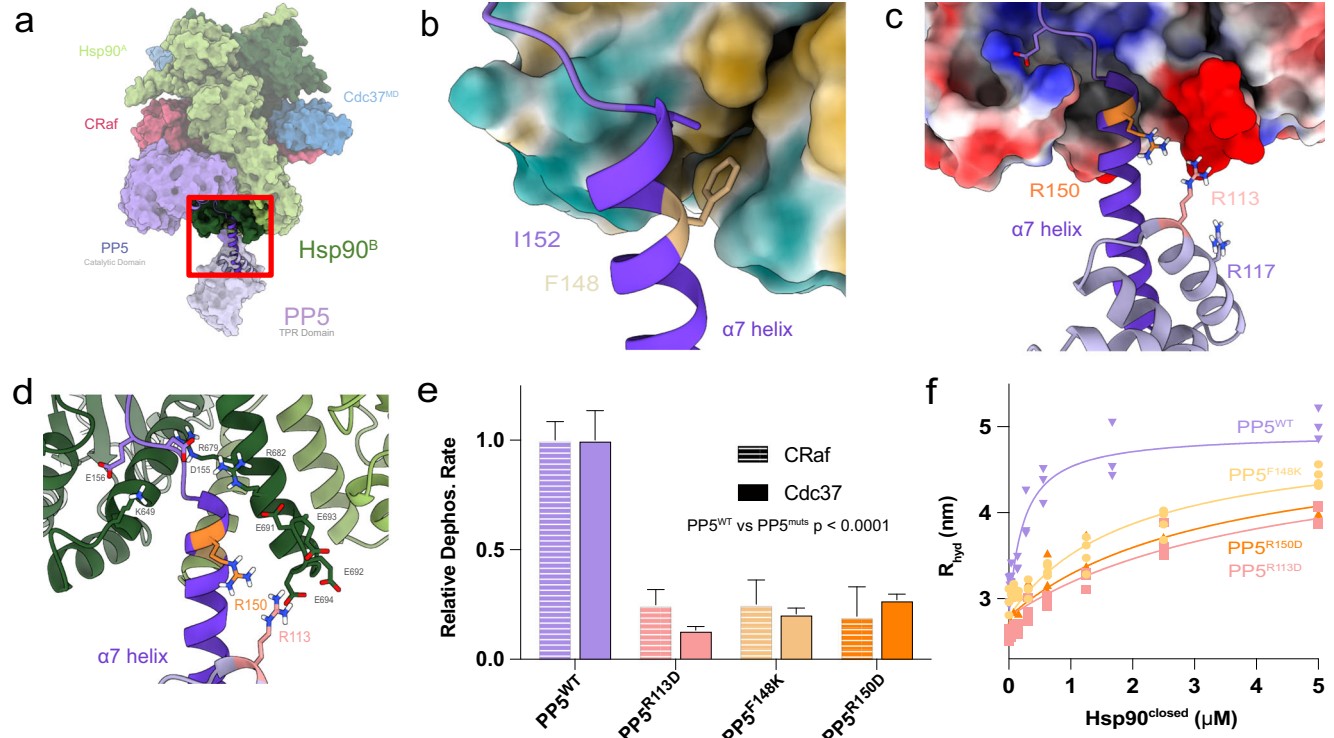

**Fig. 3 | Distinct TPR α7 helix – Hsp90 C-terminal domain interaction mode.**
**a** The PP5 TPR domain interacts with the amphipathic Hsp90 dimer C-terminal groove. **b** Hydrophobic residues PP5$^{F148}$ and PP5$^{I152}$ bind the Hsp90 hydrophobic CTD groove, while **c**, **d** PP5$^{R113}$ and PP5$^{R150}$ electrostatically interact with the Hsp90 acidic tail. **e** Mutations to key TPR residues significantly decrease PP5 dephosphorylation of both Cdc37$^{pS13}$ and CRaf$^{pS338}$ substrates (mean rates ± SEM (min$^{-1}$) reported, CRaf: PP5$^{WT}$ = 0.19 ± 0.016, PP5$^{R113D}$ = 0.046 ± 0.013, PP5$^{F148K}$ = 0.05 ± 0.02, PP5$^{R150D}$ = 0.04 ± 0.03, Cdc37: PP5$^{WT}$ = 0.043 ± 0.006, PP5$^{R113D}$ = 0.006 ± 0.001, PP5$^{F148K}$ = 0.009 ± 0.001, PP5$^{R150D}$ = 0.012 ± 0.001). The differences between PP5 mutants and PP5$^{WT}$ were significant in an ordinary one-way ANOVA (CRaf: $n$ = 3,

$p$ < 0.0001; Cdc37: $n^{mutants}$ = 3, $n^{WT}$ = 4, $p$ < 0.0001) with a Dunnett test for multiple hypothesis testing (PP5$^{WT}$ vs PP5$^{muts}$ $p$ < 0.0001). **f** Mutations to key TPR residues decrease Hsp90$^{closed}$:PP5 complex formation measured by FCS hydrodynamic radius. Hsp90$^{closed}$ dimer concentrations were used for $K_D$ evaluation. Hsp90:PP5 $K_D$'s were solved by nonlinear regression ($K_D$ ± SEM (mM) reported, $n^{mutants}$ = 4, $n^{WT}$ = 3: PP5$^{WT}$ = 0.23 ± 0.04; PP5$^{R113D}$ = 4.3 ± 0.8; PP5$^{F148K}$ = 2.0 ± 0.4; PP5$^{R150D}$ = 3.3 ± 0.6). R$_{hyd}$max and R$_{hyd}$PP5 were fit globally, an ordinary one-way ANOVA showed significant differences ($F$ = 9.8, $p$ < 0.0001) with Dunnett's multiple hypothesis test for comparison of individual interactions (PP5$^{WT}$ vs PP5$^{F148K}$ $p$ = 0.09, vs PP5$^{R113D}$ $p$ = <0.0001, vs PP5$^{R150D}$ $p$ = 0.0006). Source data are provided as a Source Data file.

Fluorescence Correlation Spectroscopy (FCS) (Fig. 3f). All three mutations led to a decrease in Hsp90$^{closed}$:PP5 complex formation, with an approximately 8-fold decrease in Hsp90$^{closed}$:PP5 binding affinity. The decrease in mutant activity and affinity confirms the broad importance of the PP5$^{TPR}$-Hsp90 interaction for substrate dephosphorylation, and suggest similar Hsp90$^{CTD}$:PP5$^{TPR}$ interfaces are used during dephosphorylation of both Hsp90$^{open}$:Cdc37$^{pS13}$ and Hsp90:Cdc37:CRaf complexes.

### Hsp90 positions the PP5 catalytic domain to efficiently dephosphorylate CRaf$^{pS621}$

While the PP5 catalytic domain is readily visible in our maps (Fig. 4a), the limited Hsp90$^{MD}$:PP5 interface size (<~330 Å$^2$) and the multiple conformations of PP5 seen through 3D local classification and 3D variability analysis (Fig. 4b and Supplementary Movie 2) suggest that the catalytic domain is only weakly stabilized by the Hsp90:kinase complex[45]. Localization is likely aided by tethering of the PP5 catalytic domain to the strongly bound TPR domain and by interactions between the catalytic domain and its CRaf substrate.

In our high-resolution structure, the PP5 active site faces toward the CRaf C-lobe. While the PP5 catalytic domain can be easily visualized, the density near the PP5 active site is quite heterogeneous. 3D variability analysis (Supplementary Movie 3) was performed on the PP5-containing particles, which led to the visualization of density extending from the C-terminal CRaf alpha helix toward the PP5 active site (Fig. 4d). Further focused classification of the CRaf:PP5 interface

revealed a class with continuous density extending from the last well resolved helical residue in the CRaf C-terminus (CRaf$^{616}$) to within 8 Å of the PP5 active site (Fig. 4e). While our sample does not demonstrate the CRaf$^{pS621}$ dephosphorylation, a plausible model was constructed based on the covalently bound PP5:substrate crystal structure (PDB: [5HPE]).

While heterogeneous density existed on the N-lobe region of CRaf, no high-resolution classes could be seen in which the PP5 catalytic domain interacted with the CRaf N-lobe. We suggest this is a consequence of using the truncated CRaf$^{KD}$ construct for our cryo-EM studies, whereas the longer CRaf$^{304–648}$ construct was used in the biochemical experiments. Thus, clear visualization of PP5 binding to the N-terminal CRaf substrate might necessitate a more stable interaction between PP5 and the CRaf N-lobe. This suggests that the residues N-terminal to the C-terminal tail may stabilize the CRaf:PP5 catalytic domain interaction, while the residues C-terminal to the CRaf$^{pS338}$ site in our construct may not be sufficient for stable PP5 interaction. To test this hypothesis, a new sample was prepared, which included the longer CRaf construct, a glutamic acid in the place of the pS338 phosphoserine, and a more activatable PP5 construct with a truncated αJ helix (see below). This Hsp90:Cdc37: CRaf$^{304–648}$:PP5$^{αJ del}$ sample was prepared as outlined previously (Supplementary Fig. 4) and a limited dataset was collected.

Although significantly fewer particles were available, 3D classification of this new dataset immediately demonstrated two distinct TPR orientations and catalytic domain density on either side of the Hsp90

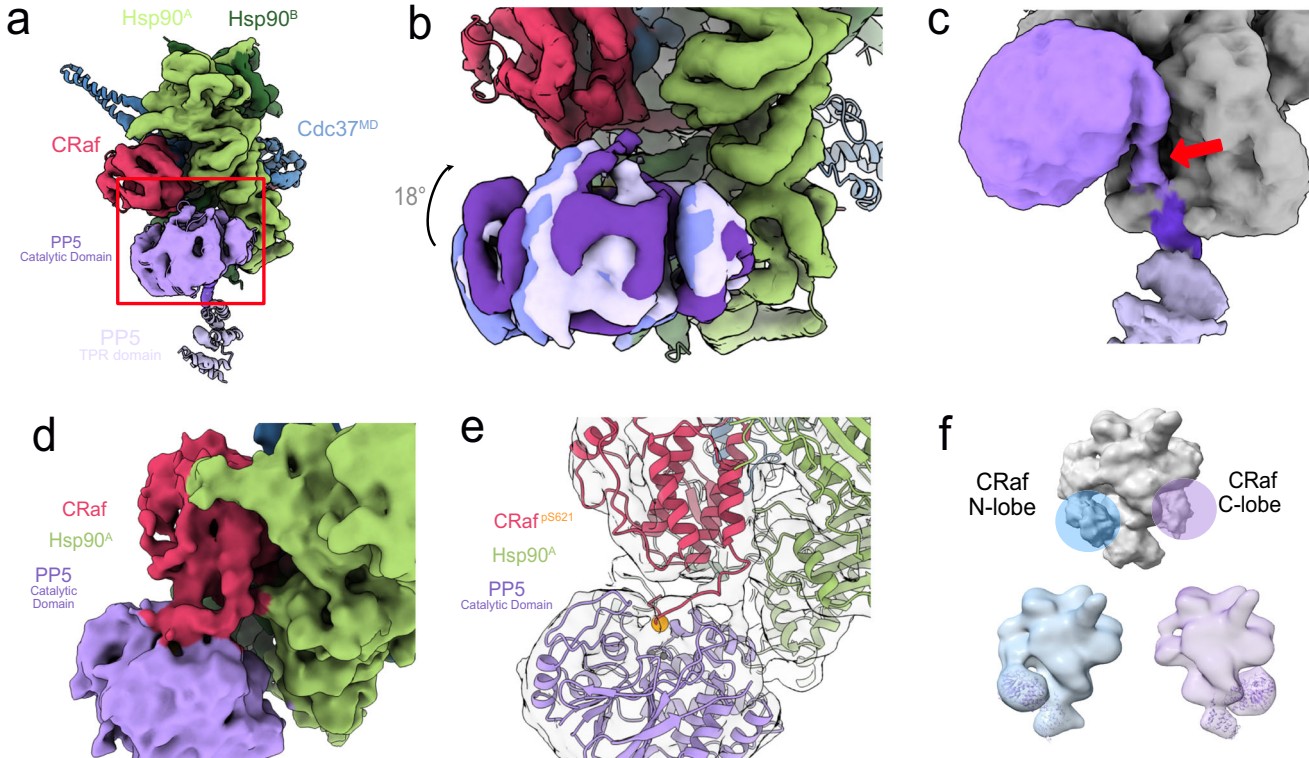

**Fig. 4 | Heterogeneous PP5 catalytic domain is in position for CRaf$^{pS621}$ dephosphorylation. a** The PP5 catalytic domain sits by Hsp90$^{MD}$ allowing for CRaf kinase dephosphorylation. **b** Multiple PP5 conformations can be teased apart through focused classification. Anchored by its TPR domain, PP5 is flexibly held in proximity to the CRaf$^{pS621}$. **c** In some classes, the PP5 linker can be seen leaving the Hsp90 CTD groove and leading up to the PP5 active site. **d** A frame from a 3D variability analysis movie (Supplementary Movie 3) shows density spanning between the C-terminus of CRaf and the PP5 active site. **e** Focused classification of the interface between the CRaf C-lobe and the catalytic domain yields a similar

class. A model was fit based on the substrate-bound PP5 crystal structure (PDB: [5HPE]) to show how CRaf$^{pS621}$ substrate may be positioned right at the PP5 active site. **f** Low-resolution volumes from the longer Hsp90:Cdc37:CRaf$^{304-648,S338E}$:PP5$^{αJ del}$ complex used in the biochemical experiments show density for the PP5 catalytic domain interacting with both the CRaf N-lobe and the CRaf C-lobe multiple PP5 orientations. The heterogeneity seen in the upper panel (gray) can be resolved into complexes having either the N-terminal or C-terminal interactions (bottom panel, purple and blue).

dimer. Further focused classification yielded low-resolution maps with the catalytic domain either in close proximity to the C-lobe as seen previously or, as predicted, in close proximity to the CRaf N-lobe (Fig. 4f).

### PP5$^{αJ helix}$ is displaced upon interaction with Hsp90:Cdc37:CRaf complex

The PP5 C-terminal αJ helix (orange) makes stabilizing interdomain interactions in the autoinhibited PP5 crystal structure (Fig. 5a). Focused classification with of the PP5 catalytic domain yielded a class of Hsp90:PP5 complexes in which no αJ helix density could be visualized (Fig. 5b). Further focused classification from this particle stack yielded one smaller class with potential low-resolution density for the αJ helix (Fig. 5b, orange). This low-resolution density also connects to the CRaf C-terminal tail and the PP5 catalytic domain. This is likely a superposition of density corresponding to an ordered αJ helix and another for the CRaf C-terminal tail. Given the small number of particles, it was not possible to further resolve this density. Previous work has suggested that the αJ helix rearranges and becomes disordered upon PP5 activation[16,46].

From these observations, we hypothesize that the removal of the αJ helix would help activate PP5. To test this, we truncated the last 10 residues of PP5 (PP5$^{αJ del}$). As shown in Fig. 5c, deletion of the αJ helix was insufficient to activate PP5 in the absence of Hsp90 (Fig. 5c). However, PP5$^{αJ del}$ did lead to a slight increase in the rates of CRaf$^{pS338}$ and Cdc37$^{pS13}$ dephosphorylation in the context of the relevant Hsp90 complexes.

To test whether this increase in dephosphorylation rate was due to a higher Hsp90:PP5 affinity, we compared the binding affinities of trace fluorescent PP5 to PP5$^{αJ del}$ bound to Hsp90$^{open}$ or Hsp90$^{closed}$. As predicted, a sharp drop in apparent $K_D$ was seen upon deleting the αJ helix, a finding which correlates well to Hsp90:PP5 and Hsp90:PP5$^{αJ del}$ sizing traces (Supplementary Fig. 9). These results suggest that the αJ helix is primarily involved in stabilizing the closed and inactive PP5 conformation, rather than being involved in Hsp90 binding. Thus, its role is to stabilize the autoinhibited state, minimizing dephosphorylation from free PP5.

## Discussion

PP5 is a unique member of the PPP family of phosphatases in that it contains a catalytic domain and a regulatory substrate-targeting TPR domain within the same protein[47]. Hsp90 offers PP5 the benefits of increased regulation and productive substrate positioning. In some cases, Hsp90 may even unfold the client, and expose phosphorylation sites that would otherwise be inaccessible.

PP5 has been shown to act as a negative regulator of numerous Hsp90 clients, including ASK1 kinase in oxidative stress, ATM kinase in DNA double-strand break stress, and Chk1 kinase in UV-induced stress[20,22,25]. Not limited to kinases, PP5 also inactivates other Hsp90 clients such as Hsf1, p53 and the glucocorticoid receptor in an Hsp90-dependent manner[21,48,49].

We set out to understand how Hsp90 activates PP5, and how Hsp90 can help position PP5 for efficient client dephosphorylation. Our Hsp90:Cdc37:CRaf$^{KD}$:PP5 complex structure illustrates the

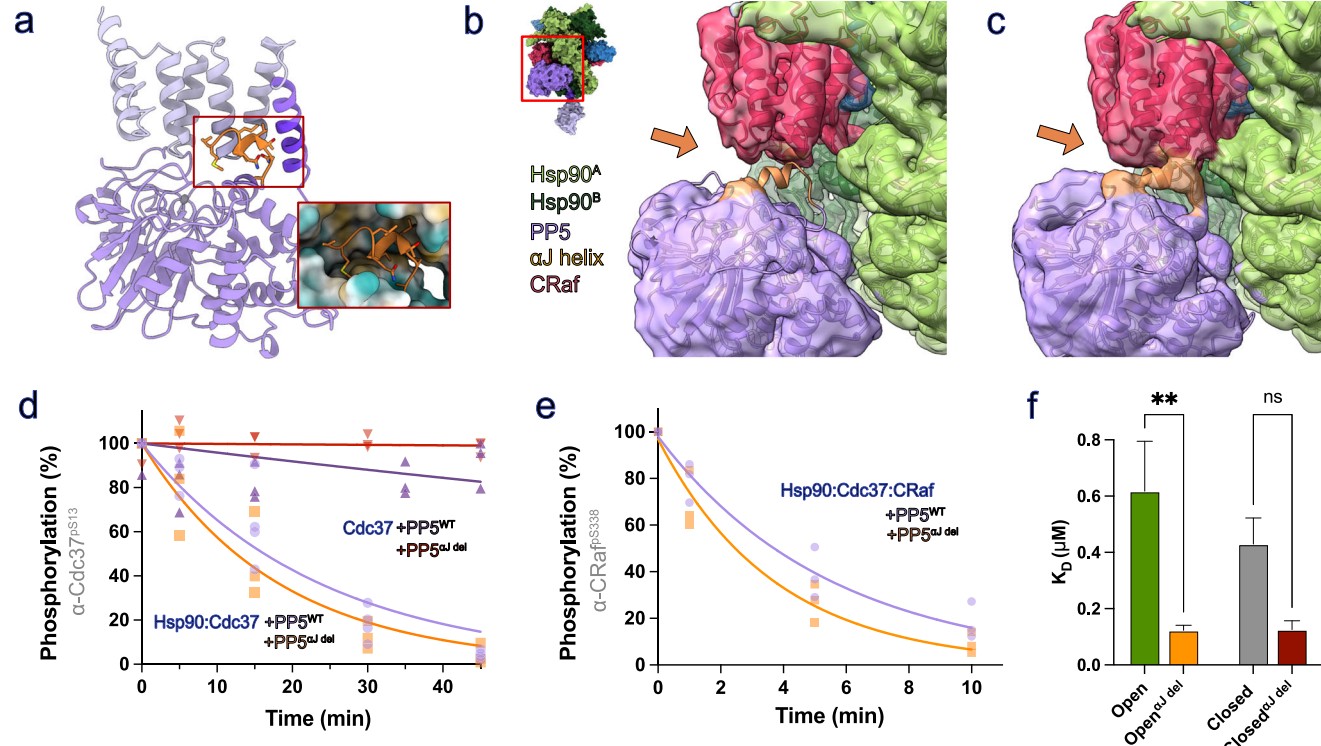

**Fig. 5 | Deletion of PP5's αJ helix increases rate of dephosphorylation. a** In its inhibited conformation, the PP5 αJ helix (orange) lies at the hydrophobic interface between the PP5 catalytic and TPR domains. **b** While initial focused classification around PP5 showed no αJ helix density, **c** further classification showed potential αJ helix density near the PP5 catalytic domain. **d** PP5$^{αJ del}$ does not dephosphorylate Cdc37 without Hsp90 present (Rates ± S.E.M. reported (min$^{-1}$), 0.0 ± 0.0), but PP5$^{αJ del}$ dephosphorylates Cdc37 while bound to Hsp90 at a slightly higher rate than PP5$^{WT}$ (0.055 ± 0.007 vs 0.043 ± 0.006). An ordinary one-way ANOVA found differences significant ($F$ = 29.40, $p < 0.0001$), while Dunnett's multiple comparisons test showed significance between Hsp90-free complexes and Hsp90:PP5$^{WT}$ complexes ($p < 0.0001$), but not between PP5$^{αJ del}$ and PP5$^{WT}$ (0.055 ± 0.007 and

0.043 ± 0.006, $p = 0.17$). **e** PP5$^{αJ del}$ led to a slight increase in CRaf$^{pS338}$ dephosphorylation as compared to PP5$^{WT}$ (0.28 ± 0.03 vs 0.187 ± 0.016 min$^{-1}$, two-tailed unpaired $t$-test: $t$ = 2.9, d.f. = 20, $p = 0.009$). **f** The binding affinity of fluorescently labeled PP5$^{WT}$ or PP5$^{αJ del}$ to Hsp90$^{open}$ or Hsp90$^{closed}$ was measured through FCS ($n$ = 3, $K_D$ ± S.E.M. ($μ$M) reported). Significant difference between the $K_D$'s of PP5$^{WT}$ and PP5$^{αJ del}$ to Hsp90$^{open}$ and Hsp90$^{closed}$ was found in an ordinary one-way ANOVA ($F$ = 5.3, $p = 0.002$), with a Šidák's multiple comparisons test comparing individual interactions (Hsp90$^{open}$:PP5$^{WT}$ vs PP5$^{αJ del}$ = 0.62 ± 0.17 vs 0.12 ± 0.018, $p = 0.0022$ and Hsp90$^{closed}$:PP5$^{WT}$ vs PP5$^{αJ del}$ = 0.43 ± 0.09 vs 0.13 ± 0.03, $p = 0.11$). Source data has been provided as a Source Data file.

molecular mechanism underlying both of these activities. The PP5 TPR and catalytic domains are separately bound to Hsp90, liberating the PP5 active site for activity and placing the phosphatase domain near its substrates. The PP5 TPR domain facilitates this by uniquely binding its α7 helix within the Hsp90$^{CTD}$ groove. While the same Hsp90 groove is used to bind to the TPR containing Fkbp51 and Fkbp52 cochaperones (Supplementary Fig. 6d–i)[37], the specialized PP5 α7 helix lies parallel to the Hsp90 dimer axis, rotated almost 90° from the α7 helix orientation seen in the Hsp90:Fkbp51/52 complexes[37,50].

The PP5 catalytic domain approaches the CRaf C-lobe, where the CRaf$^{pS621}$ phosphosite can be dephosphorylated by PP5. While the shorter CRaf$^{KD}$ likely precluded the visualization of PP5:CRaf$^{pS338}$ interactions in our structure, the use of the longer CRaf$^{304–648}$ construct in our activity assays demonstrates efficient dephosphorylation of both CRaf$^{pS338}$ and CRaf$^{pS621}$ by PP5. From this, we suggest that the catalytic domain could alternatively bind to the opposite site on the Hsp90$^B$ protomer, positioning it near the CRaf N-lobe. This hypothesis is supported by low-resolution cryo-EM densities showing that PP5 can orient itself on either face of Hsp90 and reach both ends of the scaffolded substrate (Figs. 4f and 6). Recent work by Oberoi et al. also corroborated this finding by using mass spectrometry to show PP5 dephosphorylation of a substantial number of phosphorylation sites both N-terminal and C-terminal to the BRaf kinase domain[51].

While our work clearly implicates PP5 in both CRaf$^{pS338}$-dependent inactivation, and in the dephosphorylation of CRaf's "maturity

marker", CRaf$^{pS621}$ [52], we note that PP5 only minimally alters CRaf$^{pS621}$ levels in vivo[13]. This could be due to the lack of specific inhibitory factors or more rapid re-phosphorylation of CRaf$^{pS621}$. Additional studies are clearly called for.

Surprisingly, while previous work showed that Cdc37$^{pS13}$ could be dephosphorylated by PP5[32], this was not the case in our molybdate stabilized Hsp90$^{closed}$:Cdc37:CRaf complexes. The phosphate on Cdc37$^{pS13}$ makes important Cdc37:Hsp90 stabilizing interactions in the various Hsp90$^{closed}$:Cdc37:kinase structures acquired to date[33,42]. These interactions render Cdc37$^{pS13}$ inaccessible to PP5 dephosphorylation and explain our current results. In addition, our current structure and biochemistry shows that the removal of the bound kinase, or a large rearrangement of Cdc37, would be required for PP5 to reach the phosphorylation site on Cdc37.

Previous results in which Cdc37 is dephosphorylated by PP5 may be explained by the dissociation of the complex due to longer reaction times at elevated temperatures. Indeed, in our hands, longer experimental timeframes and higher temperatures led to both Cdc37$^{pS13}$ dephosphorylation and complex dissociation (Supplementary Fig. 3).

In our experiments, kinases sterically block the dephosphorylation of Cdc37 by PP5. This negates the hypothesis that PP5 can dephosphorylate Cdc37 in an Hsp90:Cdc37:kinase complex, leading to kinase release. Instead, we propose a model in which PP5 only dephosphorylates Cdc37 once the kinase has been released, thereby resetting Cdc37$^{pS13}$ phosphorylation and preventing immediate kinase

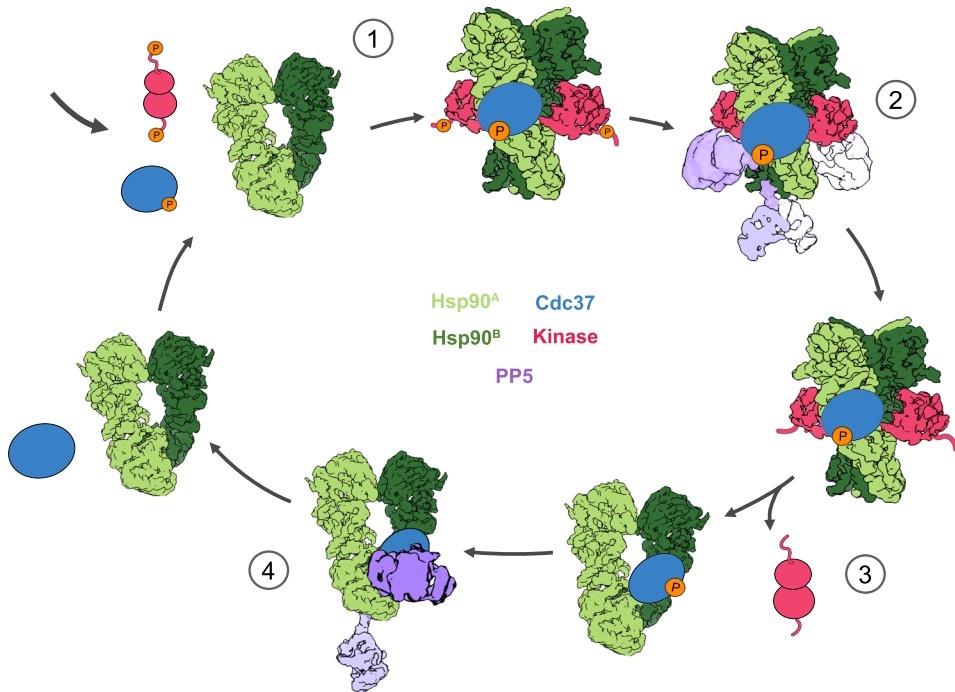

**Fig. 6 | Model of PP5's role in kinase and Cdc37 dephosphorylation.** Steric blocking of Cdc37$^{pS13}$ dephosphorylation by kinase clients suggests that Cdc37 can only be dephosphorylated once the kinase has exited the Hsp90 complex. Our model goes as follows: (1) Cdc37$^{pS13}$ first recruits the kinase to Hsp90 for folding or modification. (2) PP5 can then dephosphorylate CRaf, binding to Hsp90 in either direction to allow for the removal of phosphate groups on either side of the Hsp90-bound kinase. (3) After dephosphorylation of CRaf and potential conformational modifications by Hsp90, the phosphorylation-free kinase can be released. (4) As the kinase is released from Hsp90, Cdc37$^{pS13}$ may remain bound to Hsp90 until it's dephosphorylated by PP5. This would allow for a unidirectional Hsp90-kinase cycle.

re-recruitment (Fig. 6). This would enhance directionality to the Hsp90-kinase cycle and allow Hsp90 to release the dephosphorylated kinase in a reset "basal" phosphorylation state.

In summary, Hsp90's role in maintaining protein homeostasis goes beyond folding and activation to include facilitating client post-translational modification. Through the Hsp90:Cdc37:CRaf$^{KD}$:PP5 cryo-EM structure, we further understand how Hsp90 activates PP5 and provides a scaffold for substrate dephosphorylation. Our biochemistry efforts suggest that PP5 may directly reset kinase dephosphorylation, while influencing the recruitment of kinases to Hsp90 by Cdc37. Further work will be required to provide a more complete mechanistic description of Cdc37$^{pS13}$ dephosphorylation.

## Methods

### Inclusion and ethics statement
All research has taken place at the University of California, San Francisco (UCSF), by local UCSF researchers. No human or animal subjects were used in our study. This research does not provide a risk to the researchers. *Nature Communications* is an open-source journal, and this work will be available to all with internet access.

### Data analysis and figure preparation
All figures were assembled using Affinity Designer v1.10.5. UCSF ChimeraX v.1.2.5 was used to render the structural data included in our figures[53,54]. All graphs and charts were created using Prism v.9.3.1 (GraphPad).

The western blot data was quantified using ImageJ 1.53k[55], and moved into Prism for further normalization to a PP5-free control (100%) and analysis. The data were displayed and fit to a one-phase decay model ($Y_0 = 100\%$, Plateau = 0%) to obtain a decay rate (K min$^{-1}$). These rates were then statistically compared in Prism (ordinary one-way ANOVA test, two-tailed one-sample *t*-test and Wilcoxson test). PP5

mutant dephosphorylation rates were normalized to wildtype PP5 dephosphorylation rates to simplify reader analysis.

The CorTector SX100 FCS data analysis software Correlation Analysis (v.3.5.010722, LightEdge Technologies, Beijing, China) was used to analyze the FCS data. Data were curated, removing large fluorescent outliers from the single-molecule data. The means of the calculated radii of hydration ($R_{hyd}$) were then further analyzed using Prism v.9.3.1 (GraphPad). Triplicate data were averaged, plotted, and the $K_D$ parameter was fit using a one-site total binding model. Ordinary one-way ANOVAs were used to test for differences in $K_D$s.

### Individual component expression and purification
The Hsp90, Cdc37, PP5, and BRaf plasmids were transfected into *E. coli* BL21 cells, and plated on antibiotic agarose plates. Overnight cultures of one colony were grown in Terrific Broth media (TB media), after which 10 mL of culture were transferred into 2–6 L of TB media. Once the cultures grew to an O.D.$_{600}$ of 0.6, cells were allowed to shake (180 R.P.M.) at 16 °C for an hour, after which they were induced with 1 mM IPTG. The temperature of the cultures was then raised to 18 °C and allowed to shake overnight (160 R.P.M.). The pellets were then spun down (4500 ×g for 10 min) and frozen until protein purification.

The thawed pellets were then resuspended with Lysis Buffer (500 mM NaCl, 50 mM Tris pH 7.8, 5% Imidazole, 5% glycerol, 1 Protease Inhibitor Tablet/50 mL of pellet suspension, 0.5 mM TCEP), and the sample was sonicated in an ice bath for five cycles, 1 min/cycle, with at least a minute between cycles. The sample was then spun down at 35,000 ×g for 30 min at 4 °C. The lysate was then run through HisTrap FF (5 mL) Nickel columns (1–2 columns) at 5 mL/min. The column was loaded onto an AKTA FPLC instrument and washed with Wash Buffer (50 mM Tris pH 7.8, 250 mM NaCl, 0.5 mM TCEP, 5% glycerol, 30 mM Imidazole, PP5 buffer included 1 mM Manganese), and then eluted with

Elution Buffer (Wash Buffer with 10% glycerol and 400 mM Imidazole). The purified lysate was then concentrated down to 5–7 mL of lysate using a Centricon tube, and subsequently cleaved overnight (3C or TEV depending on plasmid).

The cleaved sample was diluted down to ~50 mL using Low Salt Buffer (20 mM Tris pH 8.0, 1 mM EDTA, 0.5 mM TCEP, 5% Glycerol, PP5 buffer included 1 mM Manganese), filtered, and loaded onto a 50 mL superloop. This sample was then run through an AKTA FPLC MonoQ 10/100 column (Cytiva), eluting through a salt gradient (0–1 M NaCl). An SDS PAGE Gel was run to select the peaks of interest, which were isolated and concentrated down to ~250 μL. The sample was then filtered and loaded onto a Superdex 200 16/600 (Cytiva) column in Storage Buffer (20 mM Hepes, 150 mM KCl, 1 mM EDTA, 1 mM TCEP, 5% glycerol). The cleanest protein fractions as dictated by an SDS PAGE Gel were concentrated and flash-frozen in liquid nitrogen for storage at −80 °C.

BRaf purification was slightly modified to improve kinase solubility: 0.75% IGEPAL was added to the Lysis Buffer, 20 mM HEPES was used instead of 20 mM Tris throughout the purification, and 10% Glycerol was used. BRaf was not cleaved overnight, and so the purification process took place in one day.

## Complex expression and purification

Hsp90:Cdc37:CRaf complexes were purified from either yeast (sample used for main Hsp90:Cdc37:CRaf:PP5 structure) or mammalian cells (sample used for biochemistry and low-resolution structure).

For yeast expression, constructs were cloned into an 83-nu yeast expression vector, and transformed into JEL1 yeast strain using Zymo Research EZ transformation protocol and plated onto SD-His plates. After 3 days, a colony was expanded into 200 mL overnight cultures. 1 L of YPGL media was inculcated with 10 mL of overnight culture. Cultures were induced with 2% w/v galactose at an O.D.$_{600}$ of 0.6–0.8 to induce expression. Temperature was reduced to 16 °C and cultures were pelleted after 18 h (4500 ×g for 5 min). Pellets were resuspended in minimal yeast resuspension buffer (20 mM HEPES-KOH pH 7.5, 150 mM KCl, 10% glycerol) and frozen dropwise into a container of liquid nitrogen.

For mammalian expression, constructs were cloned into a pcDNA3.1 expression vector. Mammalian HEK293 cells were seeded at 0.5 M/mL and allowed to reach a confluency of 3 M/mL. Media was exchanged 3 h before transfection and cells were allowed to recover. The Expi293™ Expression system kit and protocol was used for transformation. Cells were allowed to grow for 48–72 h before they were spun down at 5000 ×g, reconstituted with PBS and frozen dropwise into liquid nitrogen.

The frozen yeast or mammalian cell drops were then ground in a cryoMill for five cycles (Precool 5 min, Run 1:30 min, Cool 2 min, 10 cps rate). The samples were reconstituted with Strep Binding Buffer (20 mM HEPES, 150 mM KCl, 10 mM MgCl, 1 mM TCEP, 10% glycerol, 0.05% Tween) with NaMo (20 mM) added when purifying the more stable "closed" Hsp90 complex. The sample was loaded onto a StrepTrap HP (Cytiva, 5 mL) column at a rate of 5 mL/min and subsequently washed with 20 mL of Strep Binding Buffer in an AKTA FPLC instrument at a flow rate of 5 mL/min and then eluted with 10 mL of Elution Buffer (Strep Binding Buffer with 10 mM Desthiobiotin). The elution fractions were then concentrated down to 250 μL and loaded onto the Superdex 200 16/600 (Cytiva) column. After running the sample through the column using Storage Buffer (20 mM Hepes, 150 mM KCl, 1 mM EDTA, 1 mM TCEP, 5% glycerol), and SDS PAGE Gel was run to choose the peak to be concentrated, flash-frozen with liquid nitrogen and finally stored at −80 °C.

## Dephosphorylation assays

PP5 Dephosphorylation reactions were carried out in Reaction Buffer (20 mM Hepes, 50 mM KCl, 10 mM Mg2Cl, 1 mM TCEP, 1 mM EDTA) in PCR tubes. Buffer exchanged Hsp90:Cdc37:CRaf, Hsp90:Cdc37, Hsp90$^{Open}$ or Hsp90$^{Closed}$ complexes were placed on a 25 °C thermocycler (~1 min) before PP5 addition. The reaction began after PP5 was added and the sample was thoroughly mixed. Sample was removed from the thermocycler at each timepoint, and the reaction was quenched using SDS-DTT. Each reaction was repeated at least three distinct times with the newly prepared sample. Dephosphorylation conditions were optimized such that the PP5 concentrations used were ideal for western blot visualization. 3 μM of Cdc37$^{pS13}$ and equimolar constituents were used for Cdc37 blots, with a final addition of 750 nM of PP5$^{WT}$ or PP5$^{mutant}$. 1.5 μM of Hsp90:Cdc37:CRaf complex with 75 nM of PP5 were used for Fig. 1 experiments, while 3 μM of Hsp90:Cdc37:CRaf complex and 150 nM PP5 were used for PP5$^{mutant}$ experiments.

The samples were run on Bolt™ 4–12% Bis-Tris gels (140 V, 65 min) using the Color Protein Standard, (NEB# P7712, Broad Range (10–250 kDa)) for molecular weight differentiation. The gels were next transferred using Invitrogen's iBlot® Gel Transfer Stacks (Nitrocellulose), following the transfer kit protocol (10 min transfer) and then stained with Ponzo stain for ~5 min to ensure equal protein transfer and constant protein concentrations (Supplementary Figs. 2 and 9). The membranes were then incubated with 5% milk on a room temperature nutator for 1 h. Primary antibodies against Cdc37$^{pS13}$ (1:5000 Phospho-CDC37 (Ser13) (D8P8F) Rabbit mAb #13248), CRaf$^{pS338}$ (1:1000, # MA5-15176 Phospho-c-Raf (Ser338) Monoclonal Antibody(E.838.4)) or CRaf$^{pS621}$(1:1000, #MA5-33196 Phospho-c-Raf (Ser621) Recombinant Rabbit Monoclonal Antibody) were then added to the membrane with 5% milk and nutated overnight at 4 °C. The membrane was next washed with TBST (80 mM Tris Base, 550 mM NaCl, 1% Tween 20 (v/w)) three times, 15 min/wash. HRP Secondary antibody was then added to the membrane (1:10,000, Anti-Rabbit NA9340V GE Healthcare UK Limited) and allowed to incubate on a nutator for 1 h at RT. Next, the membrane was washed three times with TBST for 15 min/wash and exposed using the Thermo Fisher protocol and chemiluminescent agents (Pierce™ ECL Western Blotting Substrate, Catalog number: 32109).

An Azure biosystems imager was used to capture Ponceau and chemiluminescence Western Blot images. The images were then analyzed using the ImageJ software[55]. Each sample was run at least three separate times to ensure replicability. The western blots were then normalized by the phosphorylated control sample (not incubated with PP5) using the Prism software. A one-phase linear decay curve was fit to dephosphorylation vs time data, and the rates of decay were compared using an ordinary one-way ANOVA test within Prism. Multiple hypothesis testing was carried out within Prism. For visualization purposes, dephosphorylation rates for mutants were normalized to wildtype rates. All these values were then plotted with the standard error of the mean error bars.

## Hsp90:Cdc37:CRaf$^{KD}$:PP5 Cryo-EM sample preparation

Yeast purified Hsp90:Cdc37:CRaf$^{336−618}$ complex was incubated with PP5$^{H304A}$ for 30 min on ice, then brought to room temperature and mixed with 0.05% glutaraldehyde (15 min). The reaction was quenched with 50 mM Tris buffer pH 8. The sample was then filtered (PVDF 0.1 μm), and 25 μL of the sample was injected into the Ettan liquid chromatography system, where it ran through the Superdex 200 3.2/200 column (Cytiva) in running buffer (20 mM Hepes pH 7.5, 50 mM KCl, 1 mM EDTA and 1 mM TCEP). The fractions with Hsp90:Cdc37:CRaf:PP5$^{H304A}$ complex were separated from the PP5 excess, concentrated to ~300 nM and added to grids (Quantifoil R1.2/1.3, gold, covered with a monolayer of graphene oxide derivatized with amino-PEG groups) in an FEI Vitrobot chamber (3 μL of sample, 10 °C, 100% humidity, 30 s Wait Time, 3 s Blot Time, −2 Blot Force) and plunged into liquid ethane[38,39]. Frozen grids were then stored in liquid nitrogen.

## Cryo-EM data acquisition and data processing

A total of 4160 micrographs were collected using SerialEM v.3.8-beta at 105,000X magnification on a Titan Krios G3 (Thermo Fischer Scientific) with a Gatan K2 camera (0.835 Å/pix pixel size) at −0.8 to −1.8 μm defocus, 16 e⁻/pix*s, 0.025 s/frame, per frame dose of 0.57 e/A²*frame accumulating to a total dose of 69 e⁻/Å².

The micrographs were motion-corrected using UCSF Motioncor2, and their CTF was estimated using CTFFIND-4.1[56,57]. Only micrographs with a CTF fit <5 Å were kept for Gaussian blob autopicking in the cryoSPARC software (v.3.3.2). The particles were then 2D classified to remove high-resolution artifact particles, while the rest of the classes were kept for the next round of classification[41]. The particles were then exported from cryoSPARC using csparc2star[58]. The coordinates were used to re-extract particles in the RELION software (v.3.1.3), where 3D classification took place[40]. An Hsp90:Cdc37:Cdk4 C-lobe model (low pass filtered from PDB: 5FWK [PDB https://doi.org/10.2210/pdb5FWK/pdb]) was used for classification and refinement.

After one round of binned 3D classification, a particle stack of approximately 1 M Hsp90-like particles was further refined and then unbinned to yield a high-resolution Hsp90 map. Masks were created to subtract regions of interest around Hsp90, and those subtracted stacks were focused classified without alignment. Classes of interest were un-subtracted and refined, and the process was continued iteratively until no improvement in resolution was seen. Subsequently, cycles of post-processing, per particle CTF refinement and refinement were repeated until no there was no resolution improvement.

## Model building and refinement

Hsp90:Cdc37:kinase cryo-EM structures (PDB: [5FWK], unpublished D. Coutandin) and PP5 crystal structures (PDB: [1WAO], [5HPE], [1S95]) were used for model building. The PP5 linker was built using RosettaCM after the whole domain docking of both PP5 domains[59,60]. The Cdc37^MD model from PDB model [5FWK] was improved by rebuilding Cdc37 using RosettaCM and ISOLDE[61]. PhenixRefine was used after model docking/building[62], and RosettaRelax was used to model lower-resolution parts of the model. ISOLDE (v.1.2) was finally used to improve clashes, Ramachandran outliers, and rotamer fits. Different B-factor sharpened maps were used to build the models depending on different resolution areas on the map.

## FCS binding assays

E. coli purified PP5 (~50 μM) was incubated with 0.8× malemide Alexa488 dye for ~6 h at 4 °C on a nutator. The sample was then buffer exchanged using a 7MWCO Zeba™ Spin Desalting Column, and subsequently purified using a Superdex 200 3.2/300 column (Cytiva). The dilute sample was then aliquoted and flash-frozen in liquid nitrogen for future use.

The thawed sample was spun down to remove aggregates and added at a constant final concentration of ~20 nM to varying concentrations of Hsp90^open, Hsp90^closed, Hsp90:Cdc37, or Hsp90:Cdc37:CRaf^304−648. Samples were placed on microscope cover glass slides (High Precision, Deckgläser 22 × 22 mm, 170 ± 5 μm No. 1.5H) and mounted on a CorTector SX100 instrument (LightEdge Technologies, Beijing, China) equipped with a 488-nm laser. Three replicates of 10−20 10 s autocorrelation measurements per sample were recorded at room temperature. Aggregate data were discarded through curve analysis. Atto488 dye was used to calibrate the measurement volume (S). The mean of the 10−20 replicate autocorrelation measurements was then fit to an equation that accounts for single 3D diffusion and triplet dynamics, starting at autocorrelation values of 0.001 ms.

$$G(\tau) = \frac{1 - T + T^*e^{\frac{-\tau}{\tau^t}}}{1 - T} * \frac{1}{N} * \frac{1}{1 + \frac{\tau}{\tau_D}} * \frac{1}{\sqrt{1 + \frac{\tau}{\tau_D^* S^2}}} \qquad (1)$$

From these parameters, the radius of hydration ($R_{hyd}$) of each sample was calculated, and the values were plotted against the Hsp90 species concentration. Three replicates were then averaged, plotted, and fit with a one-site total binding nonlinear regression equation in Prism to obtain an approximate $K_D$ value:

$$Y = R_{hyd}Complex_{max} * \frac{X}{K_D + X} + R_{hyd}PP5 \qquad (2)$$

The background $R_{hyd}$ of PP5$_{free}$ ($R_{hyd}PP5$) value was fit globally amongst all samples, and the mutant samples were fit with the same $R_{hyd}Complex_{max}$. Different $K_D$ values were then compared to each other and assessed for the significant difference using Prism's ordinary one-way ANOVA, with Šidák's multiple comparisons to compare between Open/Closed and Open^aJ del/Closed^aJ del.

## Reporting summary

Further information on research design is available in the Nature Portfolio Reporting Summary linked to this article.

## Data availability

The raw western blot images can be found in the Supplementary figures included in this publication. The western blot and FCS data used are available in the provided Data Source file (SourceData.xlsx) included in this publication. The cryo-EM maps generated in this study were deposited in the Electron Microscopy DataBank (EMDB) and atomic coordinate models generated were deposited in the Protein Data Bank (PDB) under the following accession codes: Hsp90:Cdc37:CRaf:PP5 Composite map I: PDB 8GFT, EMDB: EMD-29984; Hsp90:Cdc37:CRaf:PP5 Composite map II: PDB 8GAE, EMDB: EMD-29895; PP5 catalytic domain, Consensus map I: EMDB: EMD-29973; PP5 catalytic domain and kinase domain, Consensus map II: EMDB: EMD-29976; PP5 TPR domain: EMDB: EMD-29957; Cdc37 Middle domain: EMDB: EMD-29949. Models used for model building and data analysis in this manuscript and in our supplementary files can be found in the PDB: 5FWK, 1WAO, 5HPE, 1S95, 6Q3Q, 7KW7, 7L7I. Source Data are provided with this paper.

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

## Acknowledgements

We thank Agard Lab members for many helpful discussions; E. Nieweglowska for her mentorship and guidance; M. Tabios for his support and encouragement; C. Nowotny, D. Mozumdar, and M. Moritz for help with western blots; T. Kortemme and J. Gestwicki for their mentorship; C. Noddings, S. Pourmal, X. Liu, and D. Asarnov for their help with Cryo-EM data processing; D. Bulkley, G. Gilbert, and Z. Yu from the W.M. Keck Foundation Advanced Microscopy Laboratory at the University of California, San Francisco (UCSF) for EM facility maintenance and help with data collection; M. Harrington and J. Baker-LePain for computational support with the UCSF Wynton Cluster. This work was supported in by NIH grant R35GM118099 (D.A.A.) and NIH grants 1S10OD026881, 1S10OD020054, and 1S10OD021741 to the UCSF cryo-EM facility.

## Author contributions

M.J.-G. designed and executed protein preparation, biochemical experiments, cryo-EM sample preparation, data collection, data processing and model building. C.A.N. helped express yeast and mammalian constructs. D.C. optimized yeast Hsp90:Cdc37:CRafKD expression and purification. F.W. optimized grids used for cryo-EM. M.T. cloned PP5 mutants. D.A.A. edited, mentored, provided funding, and gave scientific advice throughout the project.

## Competing interests

The authors declare no competing interests.
