## [Peer Review File · Nature Communications]

Hsp90 provides a platform for kinase dephosphorylation by PP5REVIEWER COMMENTS

Reviewer #1 (Remarks to the Author):

This study addresses an important aspect of the mechanism of the molecular chaperone Hsp90. One of the co-chaperones of Hsp90 is a phosphatase (PP5) and despite efforts over the years, it is still enigmatic what its role in the Hsp90 cycle is. Here the authors present the cryo EM structure of complexes containing PP5/Hsp90/Cdc37 and CRaf. The spacial organization of PP5 in the complex suggests a specific order of events in the context of the folding of kinases by Hsp90 and the dephosphorylation of the co-chaperone Cdc37. The structural work is accompanied by functional assays. Together, these results present a significant step forward in our understanding of the Hsp90 chaperone machinery.

Specific points

1. Fig 1a. shows faster dephosphorylation of residue S338 in CRaf compared to S621, however the cryoEM structure only provides information on the C-lobe position of CRaf with respect to catalytic domain of PP5 and not on the N-lobe position of CRaf. No explanation for the difference in kinetics of dephosphorylation of positions S338 compared to S621 of CRaf is provided although it is mentioned that the catalytic domain of PP5 resides closer to the latter residue of CRaf. In the discussion, the authors hypothesize a mirrored conformation for dephosphorylation of S338. It is not clear on what information this statement is based. Are their populations of complexes that would fit the mirrored conformation? Could there be other explanations for the different kinetics?
2. The authors speculate about different affinities of PP5 for open or client-loading conformations of Hsp90 and later on also about different affinities of PP5 mutants for Hsp90. However, experimental evidence is lacking.
3. Similar colors have been used to explain the interaction within domains in PP5 in Fig. 5a and for different proteins in the complex of 5b; e.g. the colors used for the α J helix of PP5 and CRaf in Fig. 5b are not very distinct.
4. The authors hypothesize that the α J helix becomes disordered upon PP5 activation. According to Fig. 5e and f, the deletion of the α J helix of PP5 leads to an increase in the dephosphorylation of CRaf and Cdc37. Did the authors try to destabilize the helix by mutation to obtain a permanently disordered helix?
5. The statement in line 162/163 “to be reported in more detail separately” is equivalent to “data not shown”. If this information is important for this study, it needs to be shown in a figure.
6. Cartoon (Fig. 6): why is PP5 not shown in complex with Hsp90 in panel B? I think this is misleading. Also, phosphorylation/dephosphorylation of the kinase is not included.

Reviewer #2 (Remarks to the Author):

In this manuscript, Jaime-Garza et. al. (Agard group) present a very elegant high-resolution structure of the quaternary Hsp90-Cdc37-CRaf-PP5 complex. The work focuses on the mechanism of cochaperone (Cdc37) and substrate (CRaf kinase) dephosphorylation while onboard to the Hsp90 chaperone. It is based on results obtained through state-of-the-art cryo-EM -aided by crosslinking of the partners to improve conformational homogeneity- and is combined with biochemical assays in support of the proposed structural data.

The most impactful highlights of the work are: (a) defining the order by which the substrate (cRaf) and cochaperone (Cdc37) dephosphorylation occurs, (b) determining the Hsp90 complex composition for these events, (c) identifying the overall Hsp90 conformation (open/closed NTD), (d) elucidating the molecular mechanism of Hsp90-induced activation of PP5, (e) providing a molecular view of active substrate dephosphorylation while Hsp90-bound is the most significant output of the study.

The manuscript is well written, and all the conclusions are nicely supported.

Overall, this is a very significant piece of work in chaperone biology. It extends the high-resolution structural work to Hsp90 complexes engaged with modifying enzymes and provides novel information on kinase processing. It deserves immediate publication at Nature Communications, after only minor revisions.

Minor comments:

1) Although dephosphorylation of pS338 is unambiguous from the biochemical assay, the manuscript would benefit if the authors further discuss -why upon crosslinking- the predominant conformation of the Hsp90-Cdc37-CRaf-PP5 complex, is such, so that the PP5 catalytic domain is positioned at the C-lobe of the kinase (in close proximity to pS621).

2) In the same lines, "Extended Fig. 10" does not seem to resolve where on the structure of the "mirrored conformation" would the PP5 catalytic domain be located, in order to achieve pS338 dephosphorylation.

3) A fair comparison to the structure presented at the bioRxiv preprint server (<https://www.biorxiv.org/content/10.1101/2022.05.03.490524v1>) will strengthen the manuscript, particularly when the authors have already performed a series of mutagenesis studies that support a novel TPR-Hsp90CTD interaction.

4) The authors should provide additional discussion on the discrepancy of pS13 dephosphorylation to (Vaughan et al Mol. Cell 2008), by further highlighting pS13 accessibility in closed complexes with a client kinase (Verba et al Science 2016 or this study).

5) Using the current structure, can the catalytic domain of PP5 reach pS13?

6) The pS13 phosphorylation differences observed between the open/closed states of Hsp90-Cdc37 complexes presented in this study is very exciting. However the term “closed” may be misleading in the absence of structural information of an actually closed binary Hsp90-Cdc37 complex.

7) pS621 prevents cRaf degradation. Can an unfruitful Hsp90 cycle benefit from PP5 dephosphorylation, where inactive cRaf (unable to autophosphorylate at S621) is degraded by the proteasome?

Reviewer #3 (Remarks to the Author):

The manuscript presented by Jaime-Garza et al shows the structure of the Hsp90:Cdc37:cRaf:PP5 complex and propose that Hsp90 activates PP5 and provides a scaffold for substrate dephosphorylation. The authors also study the dephosphorylation dynamics and suggest that the kinase is readily dephosphorylated while the Hsp90 cochaperone Cdc37 dephosphorylation is blocked until the substrate is released. The 3D reconstruction allows describing a novel interaction between Hsp90 C-terminal domain and PP5 $\alpha 7$ helix. This work constitutes one step forward to complete the knowledge on the role of Hsp90 and Cdc37 on the kinase regulation following their own previous work. However, some issues should be addressed before considering publication.

One major concern is related with the validation report provided, which is only a preliminary report that remains to be validated. The authors must provide the final report. Besides, the different protein chains must be identified to assess the confidence of the model and the described interactions.

During the revision of this manuscript, the structure of Hsp90:cRaf1:Cdc37 complex has been published (García-Alonso et al, 2022; Mol. Cell 82, 1-15). The information provided by this work should be incorporated to the present manuscript given the proximity of both complexes. Structural information in the absence of PP5 can be compared with the quaternary complex to elucidate conformational changes in the proteins.

There are other issues that should be addressed:

-The use of proteins from different origin and recombinant sources is confusing and not justified in the text. It is unclear why the authors used the yeast versions instead of the mammals. The origin and purification of PP5 are not described, nor are its mutants. Since the activity of PP5 is key for the result interpretation, detailed information must be provided.

-The PP5 mutant H304A must be explained and cited properly in the main text. It is confusing that the conformation found in the 3D reconstruction of the complex is referred to as “active” even when the mutant is not active.

-Following the previous point, it is quite unexpected that only one conformation of the quaternary complex is found. Even if the proposed steric clashes between Hsp90 C-terminal domain and PP5

happen, several intermediate conformations could be present in the sample. The heterogeneity stated by the authors could be partly due to this. I wonder if enough effort has been made to separate particles in different classes.^[1]^[SEP]

-The resolution of the 3D reconstruction is not uniform, as expected. The local resolution map showed in Sup. Fig. 4 estimates the resolution in the volume attributed to PP5 at around 5-7 Å. At that resolution some secondary structural elements should be visualized, especially the alpha helices. However, this is not the case, and there is a lack of defined elements in that density, or, at least, they are not easily appreciated with the views provided. Even though the docking of PP5 is straightforward, a lot of detailed information is extracted from it, and in some cases may lead to an overinterpretation of the structural data. For instance, I would suggest to confirm the contacts described in Fig. 4b using mutants or other techniques such as crosslinking coupled to mass spectrometry.

-Following with PP5, the authors use the docking of the crystallographic structure of isolated, autoinhibited PP5 to justify that an inactive conformation of PP5 is not possible in the complex. This extrapolation is not necessarily correct, and many changes would be expected when PP5 is bound to Hsp90, even more when visualized by EM, when several conformations can be present at the same time. Again, in my opinion, some conformations closer to the autoinhibited form should be observed and might be distinguished upon thorough classification. Alternatively, proteins from the different recombinant sources described here could be assayed too.

-The interaction mode proposed for the TPR $\alpha 7$ helix of PP5 and Hsp90 Ct is very interesting, but quite surprising too. I wonder if the interaction with the MEEVD motif is also required for a productive binding and activation, given that mutations in the putative interacting residues in $\alpha 7$ helix strongly reduce phosphorylation. Have any mutants in MEEVD motif been assayed? Sup. Fig. 6 suggests that this interaction is also happening, but the observed density is not enough to accommodate most of it, so further evidence that this peptide is actually present in the 3DR would be desirable.

-The results showing the unpaired dephosphorylation of Cdc37 when the kinase is bound are somehow contradictory with previous evidence. This doesn't mean that the observations are not correct, but the dephosphorylation model proposed based on steric impediments while the substrate is bound might need stronger support. Would it be possible to obtain a 3DR of the ternary complex Hsp90:Cdc37:PP5 that confirms the proposed changes upon the kinase release?

Other minor points:

-Many references are indicated after periods (for instance, .4). The number should be indicated before the symbol (4.).

-Fig 2: only one view of the 3D reconstruction is shown, so it is hard to assess the quality of the densities corresponding to each protein. At least the same orthogonal views depicted for the atomic model should be shown for the 3D reconstruction. The selected colours do not allow an easy visualization of Hsp90 monomers and Cdc37.

-Line 153: a space is missing, 3.3 Å.

-Fig. S9: the size and resolution of the images does not allow a proper analysis. If all these blots are shown seeking transparency of the data (which is really appreciated), they should be readable.

REVIEWER COMMENTS

Reviewer #1 (Remarks to the Author):

This study addresses an important aspect of the mechanism of the molecular chaperone Hsp90. One of the co-chaperones of Hsp90 is a phosphatase (PP5) and despite efforts over the years, it is still enigmatic what its role in the Hsp90 cycle is. Here the authors present the cryo EM structure of complexes containing PP5/Hsp90/Cdc37 and CRaf. The spatial organization of PP5 in the complex suggests a specific order of events in the context of the folding of kinases by Hsp90 and the dephosphorylation of the co-chaperone Cdc37. The structural work is accompanied by functional assays. Together, these results present a significant step forward in our understanding of the Hsp90 chaperone machinery.

Thank you!

1. Fig 1a. shows faster dephosphorylation of residue S338 in CRaf compared to S621, however the cryoEM structure only provides information on the C-lobe position of CRaf with respect to catalytic domain of PP5 and not on the N-lobe position of CRaf. No explanation for the difference in kinetics of dephosphorylation of positions S338 compared to S621 of CRaf is provided although it is mentioned that the catalytic domain of PP5 resides closer to the latter residue of CRaf. In the discussion, the authors hypothesize a mirrored conformation for dephosphorylation of S338. It is not clear on what information this statement is based. Are their populations of complexes that would fit the mirrored conformation? Could there be other explanations for the different kinetics?

Yes! We were also surprised to see PP5 bound to the CRaf C-lobe at first. The lab had previously used CryoEM to obtain structures of the Hsp90:Cdc37:Cdk4, Hsp90:Cdc37:Her2, and Hsp90:Cdc37:CRaf all using just the kinase domains. For CRaf this meant using residues 336-648, truncated two residues away from the PP5 theoretical dephosphorylation site CRaf^{fpS338}. We built on this structure by adding PP5 to this complex and collecting the large dataset used throughout these experiments.

Simultaneously, we optimized our dephosphorylation assay, where we learned that an extended CRaf construct (residues 304-648) gave stronger CRaf^{fpS338} phosphorylation. This was then used for subsequent dephosphorylation experiments, where we saw both CRaf^{fpS338} and CRaf^{fpS621} become dephosphorylated.

Through rounds of data analysis of the Hsp90:Cdc37:CRaf³³⁶⁻⁶⁴⁸ complex, it became clear that PP5 was bound to the CRaf^{fpS621} site. Although we searched for alternate PP5 conformations, none were convincing, suggesting that the two residues flanking CRaf^{fpS338} were insufficient to stabilize the PP5:CRaf^{fpS338} interaction.

We then decided to use CryoEM to look at the same complex that had been used in our biochemical data, this time with a phosphomimic at position 338, and an activating mutation on PP5 (PP5^{αJ del}). Unfortunately, even after numerous attempts, we were only able to obtain a small data set from this construct. Albeit at low resolution, this showed two orientations for the PP5 TPR domain and catalytic domain densities on either side of Hsp90. These new insights

from the low-resolution structures have now been included in Figure 4f. Despite the limited resolution, these observations clearly supported that PP5 could bind to either Hsp90 protomer, with the catalytic domain location determined in significant part by interactions with the client kinase.

2. The authors speculate about different affinities of PP5 for open or client-loading conformations of Hsp90 and later on also about different affinities of PP5 mutants for Hsp90. However, experimental evidence is lacking.

While measuring the affinity of multi-component, and multi-conformation systems has strong limitations, we shared the reviewer's curiosity and explored the binding affinity between PP5 and Hsp90 complexes. We decided to use Fluorescence Correlation Spectroscopy to track PP5's hydrodynamic radius and assess complex formation without tethering any of our highly dynamic proteins.

We were able to find approximate K_D 's between PP5^{H304A} and Hsp90^{Open}, Hsp90^{Closed}, Hsp90:Cdc37, and the Hsp90:Cdc37:CRaf^{F304-648 S338E} complexes. These different PP5:Hsp90 complexes had statistically different K_D 's as measured by an Extra sum-of-squares F-test ($p = 0.0111$) but individual comparisons were not significant because of sample noise. Because of this, this data was not included in our manuscript but is included below.

Comparison of wildtype and mutant PP5 binding to Hsp90^{closed} yielded large differences in PP5 binding. This data is included in the manuscript (Fig. 3f) and show a decrease in affinity and as a consequence a reduced plateau for the mutants. These results corroborate our structural analysis and our dephosphorylation assays.

The final comparisons done were between Hsp90:PP5^{WT} complexes and Hsp90:PP5 ^{α J del} complexes. As shown in Figure 5e, there is a significant increase in PP5 affinity when the α J helix is deleted from PP5. These results correlate well with our measured dephosphorylation data.

3. Similar colors have been used to explain the interaction within domains in PP5 in Fig. 5a and for different proteins in the complex of 5b; e.g. the colors used for the α J helix of PP5 and CRaf in Fig. 5b are not very distinct.

Thank you for the suggestion. The colors have been changed.

4. The authors hypothesize that the α J helix becomes disordered upon PP5 activation. According to Fig. 5e and f, the deletion of the α J helix of PP5 leads to an increase in the dephosphorylation of CRaf and Cdc37. Did the authors try to destabilize the helix by mutation to obtain a permanently disordered helix?

This is an interesting point, but we did not try to destabilize the helix through mutation. There are two previous crystallographic observations suggesting that the α J helix becomes disordered when the TPR domain is separated from the catalytic domain^{1,2}.

We believe that our data on the deletion of the α J helix (Figs. 5c,d,e) supports its importance in stabilizing the auto-inhibited form of PP5, but that it becomes disordered once the domains are separated.

5. The statement in line 162/163 “to be reported in more detail separately” is equivalent to “data not shown”. If this information is important for this study, it needs to be shown in a figure.

During this manuscript’s revision, a structure of Hsp90:Cdc37:CRaf was published by another lab in which this interaction was shown. This work has been cited in the place of the “to be reported in more detail separately”³.

6. Cartoon (Fig. 6): why is PP5 not shown in complex with Hsp90 in panel B? I think this is misleading. Also, phosphorylation/dephosphorylation of the kinase is not included.

This observation is well received, the model has been updated to improve clarity. Thank you.

Reviewer #2 (Remarks to the Author):

In this manuscript, Jaime-Garza et. al. (Agard group) present a very elegant high-resolution structure of the quaternary Hsp90-Cdc37-CRaf-PP5 complex. The work focuses on the mechanism of cochaperone (Cdc37) and substrate (CRaf kinase) dephosphorylation while onboard to the Hsp90 chaperone. It is based on results obtained through state-of-the-art cryo-EM -aided by crosslinking of the partners to improve conformational homogeneity- and is combined with biochemical assays in support of the proposed structural data. The most impactful highlights of the work are: (a) defining the order by which the substrate (cRaf) and cochaperone (Cdc37) dephosphorylation occurs, (b) determining the Hsp90 complex composition for these events, (c) identifying the overall Hsp90 conformation (open/closed NTD), (d) elucidating the molecular mechanism of Hsp90-induced activation of PP5, (e) providing a molecular view of active substrate dephosphorylation while Hsp90-bound is the most significant output of the study.

The manuscript is well written, and all the conclusions are nicely supported.

Overall, this is a very significant piece of work in chaperone biology. It extends the high-resolution structural work to Hsp90 complexes engaged with modifying enzymes and provides novel information on kinase processing. It deserves immediate publication at Nature Communications, after only minor revisions.

Thank you!

Minor comments:

1) Although dephosphorylation of pS338 is unambiguous from the biochemical assay, the manuscript would benefit if the authors further discuss -why upon crosslinking- the predominant conformation of the Hsp90-Cdc37-CRaf-PP5 complex, is such, so that the PP5 catalytic domain is positioned at the C-lobe of the kinase (in close proximity to pS621).

Excellent point also made by Reviewer 1. See our response above, for reviewer 1, question 1.

2) In the same lines, “Extended Fig. 10” does not seem to resolve where on the structure of the “mirrored conformation” would the PP5 catalytic domain be located, in order to achieve pS338 dephosphorylation.

Low resolution maps using the longer “extended domain” of the kinase (CRaf³⁰⁴⁻⁶⁴⁸) directly show the alternate site for PP5 (Fig. 4f). After the first round of 3D classification, we can immediately see two TPR domains bound to the Hsp90^{CTD}. This is what inspired the mirrored conformation comment on the first version of the manuscript. After submission we refined this class, and we could more clearly see two symmetric densities where PP5 might be bound.

Further focused classification yielded populations with distinct PP5 conformations. To be more precise in our language this is not a mirrored conformation, but a symmetric one. The PP5 TPR $\alpha 7$ helix appears to predominantly bind to the same Hsp90^{CTD} protomer as the PP5 catalytic domain. Fortunately, this data is sufficient to clearly demonstrate that the PP5 catalytic domain can interact with either kinase N-lobe or C-lobe sites, even if it fails to provide a detailed view of N-lobe:catalytic domain interactions. Unfortunately, despite numerous attempts, significant challenges in sample preparation have limited us to this small data set and, consequently low resolution analysis. Going beyond this point, is thus out of the scope of this manuscript.

3) A fair comparison to the structure presented at the bioRxiv preprint server (<https://www.biorxiv.org/content/10.1101/2022.05.03.490524v1>) will strengthen the manuscript, particularly when the authors have already performed a series of mutagenesis studies that support a novel TPR-Hsp90CTD interaction.

This work was wonderful to see, as it strongly affirms the interactions we see in our structure. Both of our manuscripts show structural proof of Hsp90:PP5 binding, and our work is complementary through our choice of differing PP5 activity assays.

Oberoi J. et al. solved a structure at a similar resolution to ours using BRaf, a kinase closely related to CRaf kinase used in our manuscript⁴. In our work, we both purified Hsp90:Cdc37:Kinase complexes, added PP5, and crosslinked the complex before placing it on grids. While they used BRaf^{F600E FL} kinase and expressed their complex in insect cells, we used the closely related CRaf kinase and expressed the complex in both yeast and mammalian cells. Both of our structures show similar kinase density (We both see a high-resolution kinase C-lobe and a low-resolution kinase N-lobe, with the rest of the kinase being too heterogeneous to visualize). We both see minimal rearrangement of the kinase complex upon PP5 addition, and we see similar binding of the PP5 TPR domain $\alpha 7$ helix to the Hsp90 C-terminal domain. They point out important interface residues such as F148, which we saw as well and mutated in our work. Through this we were able to demonstrate specific residue's importance in activity and

binding assays. Both of our work shows classes with the PP5 catalytic domain near the C-terminal end of the kinase C-lobe.

The largest difference in our manuscripts is in the way we chose to probe for PP5 function. We took a detailed biochemical approach, while they took a broader mass spec phosphorylation approach. By testing distinct phosphorylation sites, protein complex conformations and protein components we began to unwrap the relevance of Cdc37 dephosphorylation by PP5 and Hsp90's role as a scaffold for kinase dephosphorylation. Oberoi et al. added an excess of PP5 to their Hsp90:Cdc37:BRaf complexes (0.3uM PP5 to 0.15uM Hsp90:Cdc37:BRaf complex) and compared the phosphorylation of their samples with and without PP5 by using Mass Spectrometry. Like us, they noticed that PP5 is not necessarily specific to certain kinase phosphorylation sites. They compare the PP5 dephosphorylation activity to a "factory reset" of the bound kinase. We made sure to include their mass spec results in our manuscript to strength both our arguments.

Overall, our work is mutually supportive and nicely complementary, and we have added a segment in our manuscript suggesting this.

4) The authors should provide additional discussion on the discrepancy of pS13 dephosphorylation to (Vaughan et al Mol. Cell 2008), by further highlighting pS13 accessibility in closed complexes with a client kinase (Verba et al Science 2016 or this study).

This discrepancy has now been mentioned under results for Figure 1 for clarity (included below). Thanks for the suggestion!

"Contrary to our expectations, Cdc37^{pS13} was not measurably dephosphorylated upon PP5 addition. These results corroborate previous structural data showing that Cdc37^{pS13} is inaccessible within the complex, yet contradict previous in vitro biochemical experiments. With substantially longer incubations at 37° C (vs RT) we can observe Cdc37^{pS13} dephosphorylation. However, size exclusion analysis suggests that this is due to partial complex dissociation at 37° C (Sup. Fig 3)."

5) Using the current structure, can the catalytic domain of PP5 reach pS13?

Using the current structure, PP5 is still not close enough to dephosphorylate pS13. There would need to be a large rearrangement of the PP5 linker to allow dephosphorylation of pS13 in this "closed" complex state. Alternatively, upon ATP hydrolysis and kinase release the Hsp90 complex could rearrange to make Cdc37^{pS13} more accessible to PP5.

6) The pS13 phosphorylation differences observed between the open/closed states of Hsp90-Cdc37 complexes presented in this study is very exciting. However the term "closed" may be misleading in the absence of structural information of an actually closed binary Hsp90-Cdc37 complex.

That's a great point. Our data points towards the Hsp90 in our Hsp90:Cdc37 complex being in a semi-closed state to allow for PP5 binding (Fig. 4e, Sup. Fig. 6b), but without nucleotide we don't expect Hsp90 to be in a fully closed conformation. We reworded the

results to make sure we were clear about this, “Notably, Hsp90 bound Cdc37^{pS13} was rapidly dephosphorylated by PP5 without the presence of any nucleotide.”

7) pS621 prevents cRaf degradation. Can an unfruitful Hsp90 cycle benefit from PP5 dephosphorylation, where inactive cRaf (unable to autophosphorylate at S621) is degraded by the proteasome?

This is a very interesting question, for which much more data would be needed for us to speculate.

Reviewer #3 (Remarks to the Author):

The manuscript presented by Jaime-Garza et al shows the structure of the Hsp90:Cdc37:CRaf:PP5 complex and propose that Hsp90 activates PP5 and provides a scaffold for substrate dephosphorylation. The authors also study the dephosphorylation dynamics and suggest that the kinase is readily dephosphorylated while the Hsp90 cochaperone Cdc37 dephosphorylation is blocked until the substrate is released. The 3D reconstruction allows describing a novel interaction between Hsp90 C-terminal domain and PP5 $\alpha 7$ helix. This work constitutes one step forward to complete the knowledge on the role of Hsp90 and Cdc37 on the kinome regulation following their own previous work. However, some issues should be addressed before considering publication.

One major concern is related with the validation report provided, which is only a preliminary report that remains to be validated. The authors must provide the final report. Besides, the different protein chains must be identified to assess the confidence of the model and the described interactions.

Definitely. An essential part of the process, and something which will be completed before the final submission. Thank you.

During the revision of this manuscript, the structure of Hsp90:Raf1:Cdc37 complex has been published (García-Alonso et al, 2022; Mol. Cell 82, 1-15). The information provided by this work should be incorporated to the present manuscript given the proximity of both complexes. Structural information in the absence of PP5 can be compared with the quaternary complex to elucidate conformational changes in the proteins.

We find an RMSD of ~ 0.8 Å between the two Hsp90:Cdc37:Raf1 complex models.³ The minor differences are found on parts of CRaf kinase, the Cdc37 middle domain, and at the Hsp90 C-terminal groove. While the differences between CRaf and Cdc37 are likely simply due to protein flexibility, it seems like PP5 binding might lead to a rearrangement of the PP5:Hsp90 interface. We have included the structural comparison in Sup. Fig. 6. Thanks for the suggestion!

There are other issues that should be addressed:

-The use of proteins from different origin and recombinant sources is confusing and not

justified in the text. It is unclear why the authors used the yeast versions instead of the mammals. The origin and purification of PP5 are not described, nor are its mutants. Since the activity of PP5 is key for the result interpretation, detailed information must be provided.

Good point. We have now clarified this further in the methods. A brief explanation here:

CRaf: Previous structural experiments had been done using the yeast expressed Hsp90:Cdc37:CRaf³³⁶⁻⁶⁴⁸ complex, which contains the basic CRaf Kinase Domain. This shorter CRaf construct created a stable complex that could be purified at higher yields and had already been visualized through CryoEM in the lab.

We then decided to optimize mammalian culture expression to ensure endogenous CRaf phosphorylation for our dephosphorylation assays. We made this complex longer to include more of the CRaf linker (CRaf³⁰⁴⁻⁶⁴⁸) and realized that we could get decent Hsp90:Cdc37:CRaf complex yield from the mammalian system. At this point, structural experiments had been concluded and we were deep into data processing.

PP5: We purified PP5, and all PP5 mutants from *e Coli* BL21 cells. You can find more detail about this expression and purification in the methods section under “Individual component expression and purification”.

-The PP5 mutant H304A must be explained and cited properly in the main text. It is confusing that the conformation found in the 3D reconstruction of the complex is referred to as “active” even when the mutant is not active.

In the text, our use of “active” refers to the active conformation, and not the catalytically active phosphatase. We’ve modified our wording to clarify. Thank you.

PP5^{H304A} lacks the histidine which coordinates the fourth oxygen atom in the phosphate group and allows for subsequent dephosphorylation (M.R. Swingle et al.)^{5,6}. We tested various mutants from this paper for catalytic function and decided to use the PP5^{H304A} mutant as it was also used and tested by von Kriesheim et al. as their catalytically dead mutant.

-Following the previous point, it is quite unexpected that only one conformation of the quaternary complex is found. Even if the proposed steric clashes between Hsp90 C-terminal domain and PP5 happen, several intermediate conformations could be present in the sample. The heterogeneity stated by the authors could be partly due to this. I wonder if enough effort has been made to separate particles in different classes.

Extensive attempts to search for alternate PP5 classes in the Hsp90:Cdc37: CRaf³³⁶⁻⁶⁴⁸:PP5^{H304A} were not successful. During our data processing using the Relion software, the current conformation of PP5 jumped out from our initial 3D classification without a need for focused classification (Sup. Fig. 4). While classifying further, we found that almost 50% of the complexes had TPR bound, while only a few classes (<10%) had the catalytic domain bound (Sup. Fig.4). Further attempts to focus classify with different masks did not yield any high-resolution conformations. The PP5 conformations we did see have been included in Fig. 4b. Following your comments, we imported all of our Hsp90-containing particles into Cryosparc for reanalysis. Further classification and heterogeneous/homogeneous refinement did not yield new densities for this dataset. While surely more PP5 conformations exist in the dataset, a larger particle stack might be required to extract these volumes from the dataset.

-The resolution of the 3D reconstruction is not uniform, as expected. The local resolution map showed in Sup. Fig. 4 estimates the resolution in the volume attributed to PP5 at around 5-7 Å. At that resolution some secondary structural elements should be visualized, especially the alpha helices. However, this is not the case, and there is a lack of defined elements in that density, or, at least, they are not easily appreciated with the views provided. Even though the docking of PP5 is straightforward, a lot of detailed information is extracted from it, and in some cases may lead to an overinterpretation of the structural data. For instance, I would suggest to confirm the contacts described in Fig. 4b using mutants or other techniques such as crosslinking coupled to mass spectrometry.

Your comment is well received. We wanted to show the PP5 linker but decided to show that as a separate subfigure (Fig. 4c) to include lower threshold PP5 maps (Fig. 4a). We have also shown a new view of the complex density in Fig. 2, and we have provided some focused classification results in Fig 4b.

As per your comment on the detailed analysis of our structure, we agree with your analysis and have ensured we're not overinterpreting our structural data in our edited manuscript. Our edited manuscript shows how PP5 can inhabit distinct conformations, likely depending on the PP5 substrate for some of its interaction stability. As explained in the manuscript, the depicted Hsp90:PP5 interaction is rather weak, "the limited Hsp90MD:PP5 interface size ($< \sim 330 \text{ \AA}^2$) and the multiple conformations of PP5 seen through 3D local classification (Fig. 4b) suggest that the catalytic domain is only weakly stabilized by the Hsp90:kinase complex". Because of this, crosslinking and mass spectrometry might not yield the most exciting insights.

-Following with PP5, the authors use the docking of the crystallographic structure of isolated, autoinhibited PP5 to justify that an inactive conformation of PP5 is not possible in the complex. This extrapolation is not necessarily correct, and many changes would be expected when PP5 is bound to Hsp90, even more when visualized by EM, when several conformations can be present at the same time. Again, in my opinion, some conformations closer to the autoinhibited form should be observed and might be distinguished upon thorough classification. Alternatively, proteins from the different recombinant sources described here could be assayed too.

We did just this. Using a longer mammalian CRaf construct for complex formation (Hsp90:Cdc37:CRaf³⁰⁴⁻⁶⁴⁸), we were able to classify and see new PP5 conformations. These have been included in Fig. 4 and explained in the text. Thank you for the suggestion.

We have modified the language used when explaining the clash that occurs between the Catalytic domain of PP5 and the C-terminal domains of Hsp90. The PP5 autoinhibited conformation would still clash with Hsp90, which as the reviewer suggests would necessitate a rearrangement of TPR domain orientation or the PP5 TPR domain-Catalytic domain interface.

-The interaction mode proposed for the TPR $\alpha 7$ helix of PP5 and Hsp90 Ct is very interesting,

but quite surprising too. I wonder if the interaction with the MEEVD motif is also required for a productive binding and activation, given that mutations in the putative interacting residues in $\alpha 7$ helix strongly reduce phosphorylation. Have any mutants in MEEVD motif been assayed? Sup. Fig. 6 suggests that this interaction is also happening, but the observed density is not enough to accommodate most of it, so further evidence that this peptide is actually present in the 3DR would be desirable.

We were excited to see this as well! Earlier this year another TPR:Hsp90 interaction also showed the use of the FKPB51 $\alpha 7$ helix to interact with Hsp90 (Sup. Fig. 6), so it seems like each cochaperone TPR domain has evolved to optimize its mode of interaction with Hsp90. We believe the MEEVD peptide is still an important part of the recruitment process, but the more specific TPR:Hsp90 interaction primes the cochaperone for activity. This can be most clearly seen in work by J. Yang et al. (Fig. 6) where they show how PP5 binds more strongly to Hsp90 than to the Hsp90 C-terminal peptide¹.

We did not assay MEEVD mutants because this has been previously done by Russell, L. C. et al.⁷ In their work they used pull-downs to identify which residues were important for Hsp90:PP5 binding and identified residues K97, R101, R74, and K32. Von Kriegsheim et al. (Fig. 1c) then used one of these mutations to show Craf's dependence on Hsp90 for dephosphorylation⁶. They mutated PP5^{K97A} and consequently found a decrease in Craf dephosphorylation and Hsp90 coelution with Craf.

While these results point towards the MEEVD:TPR interaction being essential for PP5 activity, we cannot confirm that the TPR domain must remain bound to the MEEVD peptide after PP5 has bound to Hsp90 via the $\alpha 7$ helix. What we see is unaccounted density in the TPR domain where we would expect the MEEVD peptide to bind. The observed density likely does not accommodate the whole MEEVD peptide likely due to limited resolution or conformational heterogeneity. In support of that, previous NMR analysis of the PP5 TPR bound to a MEEVD peptide by M.J. Cliff et al. also concluded that there was significant heterogeneity in MEEVD binding to the PP5 TPR domain⁸.

-The results showing the unpaired dephosphorylation of Cdc37 when the kinase is bound are somehow contradictory with previous evidence. This doesn't mean that the observations are not correct, but the dephosphorylation model proposed based on steric impediments while the substrate is bound might need stronger support. Would it be possible to obtain a 3DR of the ternary complex Hsp90:Cdc37:PP5 that confirms the proposed changes upon the kinase release?

That would be very nice, and in fact is a next project, but beyond this manuscript. Unfortunately, open Hsp90 complexes have proven extraordinarily difficult to study by cryoEM due to sample heterogeneity.

The contradictory evidence also intrigued us, and we hypothesized that Cdc37 might become dephosphorylated upon Hsp90:Cdc37:Craf complex dissociation. When the Hsp90:Cdc37:Craf complex was incubated at 37°C with PP5, we could see Cdc37 become dephosphorylated (Sup. Fig. 4). This sample was subsequently run through a sizing column, after which the protein contents are analyzed through an SDS gel. As hypothesized, we found that Cdc37 was no longer bound to the larger Hsp90 complex.

Other minor points:

-Many references are indicated after periods (for instance, .4). The number should be indicated before the symbol (4.).

Great to know. All references have been corrected.

-Fig 2: only one view of the 3D reconstruction is shown, so it is hard to assess the quality of the densities corresponding to each protein. At least the same orthogonal views depicted for the atomic model should be shown for the 3D reconstruction. The selected colours do not allow an easy visualization of Hsp90 monomers and Cdc37.

The colors have been modified to allow for easier viewing, and an alternate view of the density has been provided.

-Line 153: a space is missing, 3.3 Å.

All Å units have been corrected, thanks!

-Fig. S9: the size and resolution of the images does not allow a proper analysis. If all these blots are shown seeking transparency of the data (which is really appreciated), they should be readable.

The size and resolution of these blots has been increased, hopefully they are now analyzable by the reader.

Thank you all again for helping us strengthen our science.

Sincerely,

David A. Agard

Bibliography

1. Yang, J. *et al.* Molecular basis for TPR domain-mediated regulation of protein phosphatase 5. *EMBO J* **24**, 1–10 (2005).
2. Oberoi, J. *et al.* Structural and functional basis of protein phosphatase 5 substrate specificity. *Proc. Natl. Acad. Sci. U.S.A.* **113**, 9009–9014 (2016).

3. García-Alonso, S. *et al.* Structure of the RAF1-HSP90-CDC37 complex reveals the basis of RAF1 regulation. *Molecular Cell* **82**, 3438-3452.e8 (2022).
4. Oberoi, J. *et al.* HSP90-CDC37-PP5 forms a structural platform for kinase dephosphorylation. <http://biorxiv.org/lookup/doi/10.1101/2022.05.03.490524> (2022)
doi:10.1101/2022.05.03.490524.
5. Swingle, M. R., Honkanen, R. E. & Ciszak, E. M. Structural Basis for the Catalytic Activity of Human Serine/Threonine Protein Phosphatase-5. *Journal of Biological Chemistry* **279**, 33992–33999 (2004).
6. von Kriegsheim, A., Pitt, A., Grindlay, G. J., Kolch, W. & Dhillon, A. S. Regulation of the Raf–MEK–ERK pathway by protein phosphatase 5. *Nat Cell Biol* **8**, 1011–1016 (2006).
7. Russell, L. C., Whitt, S. R., Chen, M.-S. & Chinkers, M. Identification of Conserved Residues Required for the Binding of a Tetratricopeptide Repeat Domain to Heat Shock Protein 90. *Journal of Biological Chemistry* **274**, 20060–20063 (1999).
8. Cliff, M. J., Harris, R., Barford, D., Ladbury, J. E. & Williams, M. A. Conformational Diversity in the TPR Domain-Mediated Interaction of Protein Phosphatase 5 with Hsp90. *Structure* **14**, 415–426 (2006).

REVIEWERS' COMMENTS

Reviewer #1 (Remarks to the Author):

The authors have carefully considered the reviewer's comments.

The revisions and new data added answer the queries in a highly satisfactory manner.

A very convincing study.

Reviewer #2 (Remarks to the Author):

The further refinement of classes that show multiple PP5 interactions (both N- and C- lobes) is a great addition to this work.

Recommendation: ACCEPT

Reviewer #3 (Remarks to the Author):

The revised version of the manuscript presented by Jaime-Garza et al. has successfully addressed most of my concerns and the response to most comments is satisfactory. I acknowledge the efforts made to improve all the issues that were remarked in the first revision. I must insist that a preliminary validation report is not intended for manuscript review and that complicates the evaluation of the quality of the model. I would just recommend to have a second look at the methods section to increase its uniformity regarding units, abbreviations, etc.

RESPONSES TO REVIEWERS' COMMENTS

Reviewer #1 (Remarks to the Author):

The authors have carefully considered the reviewer's comments. The revisions and new data added answer the queries in a highly satisfactory manner. A very convincing study.

Thank you so much for your feedback and support.

Reviewer #2 (Remarks to the Author):

The further refinement of classes that show multiple PP5 interactions (both N- and C- lobes) is a great addition to this work.
Recommendation: ACCEPT

Thank you, it's exciting data!

Reviewer #3 (Remarks to the Author):

The revised version of the manuscript presented by Jaime-Garza et al. has successfully addressed most of my concerns and the response to most comments is satisfactory. I acknowledge the efforts made to improve all the issues that were remarked in the first revision. I must insist that a preliminary validation report is not intended for manuscript review and that complicates the evaluation of the quality of the model. I would just recommend to have a second look at the methods section to increase its uniformity regarding units, abbreviations, etc.

Thank you for your suggestions and time.

The submission to the PDB has been completed, the validation for manuscript review is attached.

Thanks, we have indeed looked closer at the methods section and improved its uniformity.